

# The skull of the Turks and Caicos rock iguana, *Cyclura carinata* (Squamata: Iguanidae)

Chloe Lai[1] and Simon G. Scarpetta[1,2]

[1] Museum of Vertebrate Zoology, University of California, Berkeley, Califiornia, United States of America
[2] Department of Environmental Science, University of San Francisco, San Francisco, CA, United States of America

## ABSTRACT

We provide a detailed and first description of the skull, hyoid apparatus, and trachea of the Turks and Caicos rock iguana, *Cyclura carinata* (Squamata: Iguanidae). *Cyclura* is a radiation of iguanas restricted to islands of the Caribbean Sea. Species of *Cyclura* have high rates of endemism, and all species are severely threatened with extinction. Our anatomical description of this threatened iguana is based on high-resolution computed tomography scans of one adult, one putative adult or near adult, and one juvenile specimen, and includes three-dimensional segmented renderings and visualizations. We discuss some observations of intraspecific and ontogenetic variation, and provide a brief comparison with specimens of another species of *Cyclura* and published descriptions of other iguanas. Our study provides a cranial osteological framework for *Cyclura* and augments the body of knowledge on iguana anatomy generally. Finally, we posit that our description and future studies may facilitate identification of fossil *Cyclura*, which could help understand the paleobiogeography of the genus.

## INTRODUCTION

Morphological data from the vertebrate skeleton provide a wealth of information in biology and paleontology. Osteological data are used to understand ecomorphology (*Watanabe et al., 2019*), species delimitation and identification (*Conrad & Norell, 2010*; *Keogh et al., 2008*), phylogeny (*Gauthier et al., 2012*; *Simões et al., 2018*), and pathology (*Rothschild, Schultze, Pellegrini, 2012*). Skeletal data also inform on adaptation, for example, the capacity of *Anolis* lizards to adapt to newly colonized environments or anthropogenic climate change (*Winchell et al., 2023*).

Lizards and snakes (Squamata) compose an exceptionally diverse vertebrate radiation containing over 11,000 extant species (*Uetz & Hallerman, 2023*). Despite the important role of skeletal data in biological research, there is a scarcity of documented skeletal data for many squamates (*Bell & Mead, 2014*; *Evans, 2008*). In particular, patterns of intraspecific variation are not well understood for many species, and there are also species, especially those that are rare and/or threatened, for which any detailed skeletal information is

Corresponding author
Chloe Lai, chloeemme@berkeley.edu

deficit or lacking entirely (*Bell & Mead, 2014*; *Villa et al., 2017*). Documenting intra and interspecific variation is necessary for a comprehensive understanding of species ecology, morphology, and for identifying fossils (*Ledesma, Scarpetta & Bell, 2021*; *Ledesma et al., 2023*). Data from the skull are particularly important, because cranial morphology contains unique ecological information (*e.g.*, cranial kinesis, diet). In lizards, osteological traits and/or patterns of extant species, such as tooth morphology or modular diversification of the skull, can be compared to fossils of extinct species to infer ecological traits such as habitat and diet (*Montanucci, 1968*; *Herrera-Flores, Stubbs & Benten, 2021*; *Watanabe et al., 2019*). Fossil skull elements are generally the most common squamate bones identified by paleontologists, especially for lizards (*Bell & Mead, 2014*), and thus understanding the cranial osteology of extant species is crucial for identifying extinct taxa.

Iguanas and chuckwallas (Squamata: Iguanidae) are a group of large and generally herbivorous lizards that inhabit North, Central, and South America, the Galápagos Islands, the Caribbean islands, and the South Pacific islands of Fiji and Tonga (*Buckley et al., 2016*). Because of their large size, recognizability, and diversity of adaptations, iguanids are marquee components of the ecosystems they inhabit. The osteology of iguanid lizards has been studied previously, often in a systematic context (*Avery & Tanner, 1971*; *Conrad & Norell, 2010*; *de Queiroz, 1987*; *González Rodríguez et al., 2023*; *Oelrich, 1956*; *Paparella & Caldwell, 2021*). The cranial osteology of iguanids has also been examined in studies of the more inclusive clade Iguania, a group that includes iguanas, horned lizards, agamas, and chameleons (*Townsend et al., 2011*). Previous work on iguanian cranial osteology spans anatomical, phylogenetic, and paleontological research (*e.g.*, *Conrad, Rieppel & Grande, 2007*; *Daza et al., 2012*; *McGuire, 1996*; *Moody, 1980*; *Scarpetta, 2024*; *Smith, 2009*).

*Cyclura* (*Harlan, 1824*), often called cycluras or rock iguanas, are especially robust, terrestrial lizards that are restricted to the Greater Antillean islands of the Caribbean Sea (*Iverson, 1979*). Sympatry among the ten presently recognized living species is uncommon, and each species for the most part solely occupies its own island and/or island bank (*Buckley et al., 2016*). In part due to their restricted range size, hunting, and collection for the pet trade (*Alberts, 2004*; *Buckley et al., 2016*; *Reynolds et al., 2022*), *Cyclura* are among the most endangered lizards on the planet. All species have a minimum IUCN Red List status of Vulnerable, many species and subspecies are Endangered or Critically Endangered (*Buckley et al., 2016*; *IUCN, 2023*), and there is one described extinct species (*Powell, 2000*). Research and especially conservation efforts have greatly increased during the 21st century, but many basic aspects of the biology of *Cyclura* are yet unknown. Specifically, few efforts have been made to document the skull anatomy of *Cyclura* (*e.g.*, *González Rodríguez et al., 2023*) and none have been made to describe and figure in detail the anatomical features of each cranial element. Many fossils of *Cyclura* have been collected from the Greater Antilles, and a better understanding of the cranial osteology of extant species would be highly useful for rigorous identifications of these fossils. Known fossils of *Cyclura* include cranial and appendicular elements (*Etheridge, 1965*; *Olson, Pregill & Hilgartner, 1990*; *Pregill, 1982*) and even nesting trace fossils (*Martin et al., 2020*) from the Bahamas, and cranial and potentially appendicular fossils from Turks and Caicos and the Cayman Islands (*O'Day,*

*2002*; *Morgan & Albury, 2013*). Most fossils of *Cyclura* were reported in faunal lists rather than formal descriptions.

Here, we provide the first complete, detailed assessment and visualization of the skull, hyoid, and trachea of any species of *Cyclura*. We leveraged high resolution X-ray computed-tomography scans (CT) to visualize the skull, hyoid, and partial trachea of our focal species, *Cyclura carinata* (Turks and Caicos rock iguana). The trachea has not been documented or visualized in most *Cyclura*, but was recently figured for *Cyclura cornuta* (*González Rodríguez et al., 2023*). The hyoid apparatus has been discussed taxonomically (*Conrad, 2008*) and described in detail for several iguanas, including the Galápagos marine iguana *Amblyrhynchus cristatus* (*Paparella & Caldwell, 2021*) and the spinytail iguana *Ctenosaura pectinata* (*Oelrich, 1956*).

The focal species, *Cyclura carinata*, is restricted to the Turks and Caicos island bank and is currently considered Endangered by the IUCN Red list (*Gerber, Colosimo & Grant, 2020*). The total population of the species is roughly 30,000 (*Gerber, Colosimo & Grant, 2020*). *Cyclura carinata* is one of the smaller members of the genus (∼35 cm snout to vent length), which has allowed them to persist on relatively small island cays but has also increased their vulnerability to invasive mammalian predators (*Gerber, Colosimo & Grant, 2020*).

We base our anatomical descriptions, images, and analyses on three specimens, so our description is fairly preliminary. That said, we provide some ontogenetic data and detailed and high-resolution figures and insights into previously undescribed morphological features of *Cyclura* (*e.g.*, the trachea). This work will provide the foundation and an anatomical reference for future investigations into the osteology and fossil record of *Cyclura*.

## MATERIALS & METHODS

### Anatomical terminology
We used the terminology of *Evans (2008)* except where otherwise noted.

### Specimens examined
Specimens examined *via* computed tomography: *Cyclura carinata*: UF Herp 32820, UMMZ 117401 (juvenile), MVZ Herp 81381. CT data for UF Herp 32820 and the juvenile UMMZ 117401 were obtained from Morphosource.org and digitally processed to provide images anatomical descriptions of the skull and the specimen MVZ Herp 81381 was CT-scanned for this project. Traditionally prepared skeletal specimens examined for comparison include *Cyclura cornuta* MVZ Herp 95982 and MVZ Herp 95983. Ontogenetic differences are discussed in the specific bone descriptions and interspecific differences are discussed in the 'Discussion' section.

### Ontogeny of specimens
Complete fusion of the braincase was previously found to indicate sexual maturity in the desert iguana *Dipsosaurus* (*Maisano, 2001*). The examined CT specimen UF Herp 32820 has near complete fusion of the braincase elements (described below), indicating sexual

maturity. MVZ Herp 81381 has fusion of most braincase elements so it is likely an adult or near adult, though some sutures can be seen (in particular between the prootic and the sphenoid). UMMZ 117401 is a juvenile specimen lacking fusion of any braincase elements.

## CT data and segmentation

Specimen and scanning information are in Table 1 and anatomical abbreviations are in Table 2. We digitally segmented and disarticulated the skulls of these specimens with the software SlicerMorph in 3D Slicer (*Fedorov et al., 2012*; *Rolfe et al., 2021*) to generate digital 3D models. Segmentation used the threshold tool with greyscale bounds to capture bone, the draw tool to select large portions of bones on the individual CT slices, and the spherical paint tool for greater precision with smaller or articulating elements. We created separate models of each segmented skull bone for comparison. Highly fused cranial elements were not fully segmented and relative position and contact between fused elements were based on observations from existing knowledge of closely related taxa. This includes a posterior fusion of the right surangular and articular, as well as the single braincase element in UF Herp 32820. The anatomical figures were created using screenshots taken in 3D Slicer of the 3D surface renderings of the skull with a smooth factor of 2.5.

## RESULTS

### General features of the skull

*Cyclura carinata* is a relatively small *Cyclura* lizard (~35 cm snout-to-vent length, ~80cm total length; *Gerber, Colosimo & Grant, 2020*). The species possesses typical iguanian, pleurodontan, and iguanid features, such as a a dorsal process of the squamosal (an apomorphy of Iguania), two foramina on the anterior process of the maxilla, including an anterior superior alveolar foramen and a subnarial arterial foramen (an apomorphy of Pleurodonta), and a parietal foramen that is at the frontoparietal suture but mainly contained by the frontal (an apomorphy of Iguanidae) (*Etheridge & de Queiroz, 1988*; *Smith, 2009*). Bones are robust and well-ossified, but there is little to no rugose sculpturing on the lateral and/or dorsal surface of any cranial element (Figs. 1 and 2). Some iguanids (*e.g.*, *Conolophus*) have more pronounced rugosities on their bones (*Paparella & Caldwell, 2021*). The orbit is large and the snout is fairly elongate (Fig. 2). Most of the teeth, especially distal teeth, are multicuspid, as in most other iguanids (*de Queiroz, 1987*; *Etheridge & de Queiroz, 1988*; *Smith, 2009*).

### Mandible

The mandible consists of the dentary, coronoid, splenial, angular, surangular, and articular (Figs. 3 and 4). Some of these bones are fused, almost indistinguishably (*e.g.*, the surangular and articular), whereas others retain clear sutural boundaries. The mandible bones of the juvenile were not distinguishable from each other, though this could have resulted from scan quality.

#### *Dentary*

The dentary is a long, robust bone located in the anterior portion of the lower jaw (Figs. 4 and 5). The posterior end is taller than the anterior end and has three separate processes.
**Table 1  Raw CT data sources and parameter settings for *Cyclura carinata*.**

| Specimen | Date and locality | Scanner | Facility | CT slices in XY plane | X-ray settings | Voxel size (mm) |
|---|---|---|---|---|---|---|
| UF Herp 32820 | 1974, Turks and Caicos, Pine Cay | General electric phoenix c\|tome\|x m240 scanner | UF Nanoscale Research Facility | 6,013 | 100 kV, 0.25 mA | 0.111482 |
| UMMZ 117401 | 1,953, Turks and Caicos, Long Cay | Nikon Metrology XT H 225 ST | UMMZ Research Museums Center | 1,908 | 85 kV, 0.2 mA | 0.076494 |
| MVZ 81381 | 1964, Turks and Caicos, Turks Island | General Electric phoenix nanotom m 180 | UCB FAVE Lab | 3,000 | 70 kV, 0.220 mA | 0.036739 |

The three processes articulate with the coronoid, articular, and surangular. The medial face of the surangular process (posterolateral process) articulates with the surangular, while the angular process (posteroventral process) contacts the angular along the ventral margin. (Figs. 5A and 5B). The shallow facet that separates the posterodorsal and surangular processes contacts the lateral process of the coronoid laterally; the lateral process of the coronoid lays on the coronoid facet of the dentary (Fig. 4C).

When the jaw is closed, dentary teeth stand just medial to the maxillary teeth. The Meckelian fossa of the dentary is fused for much of its length and opens anteriorly, ventral to the mandibular symphysis (Fig. 5A). Posteriorly, it is filled by the splenial and coronoid. The anterior inferior alveolar foramen is ventrally bordered by the splenial and dorsally bordered by the subdental shelf of the dentary (Figs. 4A and 5B). There is no distal subdental shelf.

Large nutrient foramina appear laterally on the anterior half of the dentary and pierce the Meckelian fossa. The right dentary has seven nutrient foramina while the left dentary has four larger nutrient foramina (Fig. 5C). The juvenile *Cyclura carinata* UMMZ 117401 is missing foramina along the lateral face of the dentary.

### *Coronoid*

The coronoid is a tetraradiate bone consisting of three processes that articulate with the mandible and a large coronoid eminence (dorsal process). The coronoid contacts the dentary anteriorly, the splenial anteroventrally, and the fused articular and surangular posteroventrally (Figs. 4 and 4A).

The tall, thin coronoid eminence is mediolaterally compressed as the body tapers to a round dorsal surface (Fig. 6B). It projects dorsally and slightly angles posteriorly (Fig. 6A). The process contacts the surangular posteroventrally, and clasps the posterior end of the dentary as the ventral coronoid process slots into the coronoid facet of the dentary. The long anteromedial process extends anteriorly to articulate dorsally with the dentary and ventrally with the articular, surangular, and splenial. The process extends anteriorly and ventral to the posterior end of the subdental shelf, reaching the third to last tooth position on the dentary (Fig. 6A). The anterior process narrows as it enters the posterior opening of the Meckelian fossa. The long posteromedial process is bifurcated and extends ventrally

**Table 2** Anatomical abbreviations.

| Abbreviation | Structure |
|---|---|
| 1cb | First ceratobranchial |
| 1eb | First epibranchial |
| 2cb | Second ceratobranchial |
| a.ar | Anterior ampullar recess |
| a.ia.f | Anterior inferior alveolar foramen |
| a.m.fo | Anterior mylohyoid foramen |
| a.Mk.fs | Anterior opening of Meckelian fossa |
| a.San.f | Anterior surangular foramen |
| a.vc | Anterior vidian canal opening |
| ab.f | Abducens foramen |
| ad.cr | Adductor crest |
| ad.fs | Adductor fossa |
| aip | Anterior inferior process |
| al.pr | Anterolateral process |
| al.Px.pr | Anterolateral premaxillary process |
| alv.p | Alveolar plate |
| am.pr | Anteromedial process |
| am.pr | Anteromedial process |
| am.Px.pr | Anteromedial premaxillary process |
| An | Angular |
| an.pr | Angular process |
| Art | Articular |
| bh | Basihyal |
| Bo | Basioccipital |
| Bo.co | Basioccipital condyle |
| Bt.pr | Basipterygoid process |
| c.co | Central column |
| cch | Conch |
| ce.co | Cephalic condyle |
| Co | Coronoid |
| co.am.pr | Anteromedial process of coronoid |
| co.em | Coronoid eminence |
| co.ft | Coronoid facet |
| co.pm.pr | Posteromedial process of coronoid |
| cr | Crest |
| cr.cr | Crista cranii |
| cr.if | Crista interfenestralis |
| cr.Pro | Crista prootica |
| cr.s | Crista sellaris |
| cr.tb | Crista tuberalis |
| crt | Crista transversalis |

| Abbreviation | Structure |
| --- | --- |
| D | Dentary |
| d.s | Dorsum sella |
| Ec | Ectopterygoid |
| eh | Epihyal |
| et.f | Ethmoidal foramen |
| f.12 | Foramina for hypoglossal nerve |
| f.7 | Foramen for facial nerve |
| f.8 | Foramen for vestibulocochlear nerve |
| f.co | Fossa columella |
| f.mg | Foramen magnum |
| f.o | Fenestra ovalis |
| fe | Free epibranchial |
| Fr | Frontal |
| Fr.n.ft | Nasal facet of frontal |
| hc | Hyoid cornu |
| Hy | Hyoid |
| i.o.f | Infraorbital foramen |
| i.pro | Incisura prootica |
| icf | Internal carotid foramen |
| J | Jugal |
| J.p.pr | Posterior process of jugal |
| L | Lacrimal |
| L.f | Lacrimal foramen |
| l.pr | Lateral process |
| L.pv.pr | Lacrimal posteroventral process |
| lhv.n | Notch for the lateral head vein |
| lrst | Lateral aperture for the recessus scali tympani |
| ma.co | Mandibular condyle |
| ma.co.ft | Mandibular condyle facet |
| ma.sy | Mandibular symphysis |
| mrst | Medial aperture for the recessus scali tympani |
| Mx | Maxilla |
| Mx.f.pr | Maxillary facial process |
| Mx.j.pr | Jugal process of maxilla |
| Mx.pr | Maxillary process |
| Mx.sh | Maxillary shelf |
| N | Nasal |
| N.al.pr | Anterolateral process of nasal |
| N.am.pr | Anteromedial process of nasal |
| n.f | Nutrient foramen |
| N.fr.pr | Frontal process of nasal |
| nu.fo | Nucal fossa |
| o.pr | Orbital process |
| Os | Orbitosphenoid |

| Abbreviation | Structure |
|---|---|
| Ot | Otooccipital |
| P | Parietal |
| P.a.pr | Parietal anterior process |
| P.f | Parietal foramen |
| P.fo | Parietal fossa |
| p.m.fo | Posterior mylohyoid foramen |
| p.San.f | Posterior surangular foramen |
| p.ta | Parietal table |
| p.vc | Posterior opening of vidian canal |
| pa.pr | Palatal process |
| Pa.pr | Palatine process |
| Pfr | Postfrontal |
| Pfr.ft | Postfrontal facet |
| pit.fo | Pituitary fossa |
| pl | Processus lingualis |
| Po | Postorbital |
| Po.pr | Postorbital process |
| poc.pr | Paraoccipital process |
| pp.pr | Postparietal (supratemporal) process |
| pr.as | Processus ascendens |
| pre.Art.pr | Prearticular process |
| Prf | Prefrontal |
| Prf.am.pr | Anteromedial process of prefrontal |
| Prf.pv.pr | Posteroventral process of prefrontal |
| Pro | Prootic |
| Pro.a.pr | Anterior process of prootic |
| psp.pr | Parasphenoid process |
| Pt.lm | Pterygoid lamina |
| Pt.t | Pterygoid teeth |
| Px | Premaxilla |
| Px.a.f | Premaxilla anterior foramen |
| Px.n.pr | Premaxillary nasal process |
| Px.pr | Premaxillary process |
| Q | Quadrate |
| Q.pr | Quadrate process |
| r.v.j | Recessus vena jugularis |
| ra.pr | Retroarticular process |
| s.t | Sella turcica |
| sa.f | Subnarial arterial foramen |
| sac | Superior alveolar canal |
| San | Surangular |
| san.pr | Surangular process |
| sd.sh | Subdentary shelf |

**Table 2** (*continued*)

| Abbreviation | Structure |
| --- | --- |
| So | Supraoccipital |
| sof | Suborbital fenestra |
| Sp | Sphenoid |
| Sp | Sphenoid |
| Sp.a.pr | Anterior process of sphenoid |
| Sq | Squamosal |
| St | Supratemporal |
| st.pr | Supratrigeminal process |
| Stp | Stapes |
| Sx | Septomaxilla |
| Sx.a.pr | Septomaxilla anterior process |
| Sx.d.pr | Septomaxilla dorsal process |
| Sx.v.pr | Septomaxilla ventral process |
| t.cr | Tympanic crest |
| Ta | Trachea |
| tr | Trabecula |
| v.ch | Vomer choana |
| v.f | Vagus foramen |
| vl.ft | Ventrolateral facet |
| vl.Pt.fl | Ventrolateral pterygoid flange |
| vm.pr | Ventromedial process |
| vm.Pt.fl | Ventromedial pterygoid flange |
| Vo.pr | Vomerine process |

and posteriorly to articulate with the facets on the articular and surangular, respectively (Fig. 6A). The posterior process splits into two relatively big projections and one small, middle projection.

A prominent adductor crest runs along the posteromedial margin of the coronoid process to the ventral end of the posteromedial process. The crest slightly flattens as it reaches the end of the ventral process (Fig. 6A). The coronoid bears two shallow foramina located dorsomedially on the posteromedial process and medial to the adductor crest.

Unlike the adult, the juvenile does not have well-developed processes of the coronoid. The anteromedial process of the juvenile is shorter than that of the adult, and does not reach the last tooth position of the dentary. The juvenile coronoid eminence contacts the ectopterygoid anteromedially and does not bear any foramina.

### Splenial

The splenial is a thin and flat bone that extends anteriorly into the Meckelian fossa. The splenial articulates with the dentary, angular, coronoid, and articular bones. The bone extends posteriorly to the midpoint of the coronoid and extends anteriorly to reach just below tooth position 14 of the dentary (Fig. 4A). The anterior process of the splenial runs under the subdental shelf, tapering to a sharp tip. The anterior end of the bone contributes to the ventral border of the anterior inferior alveolar foramen. The anterior

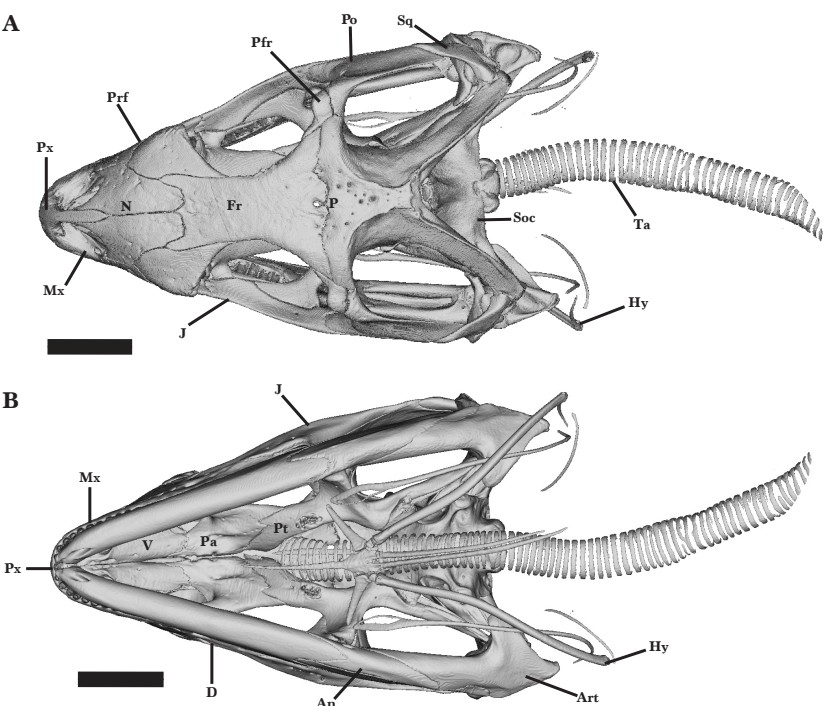

**Figure 1 Full skull dorsal and ventral view.** Entire skull of *Cyclura carinata* UF:Herp:32820 (ark:/87602/m4/M59620). (A) Dorsal view, (B) ventral view. Scale bar = 10 mm.

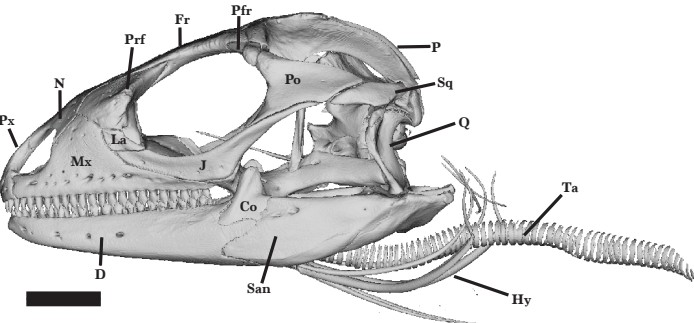

**Figure 2 Reference skull lateral view.** Lateral view of the entire skull of *Cyclura carinata* UF:Herp:32820 (ark:/87602/m4/M59620). Scale bar = 10 mm.

mylohyoid foramen pierces the splenial posteroventral to the anterior inferior alveolar foramen (Fig. 4A). The anterior mylohyoid foramen is approximately half the size of the anterior inferior alveolar foramen and more oval-shaped.

The splenial expands posteriorly into two processes. The posteroventral process is much longer than the posterodorsal process and fits in between the angular and articular. It tapers dorsoventrally to a sharp tip. The posterodorsal process slots into a facet of the anteromedial process of the coronoid.

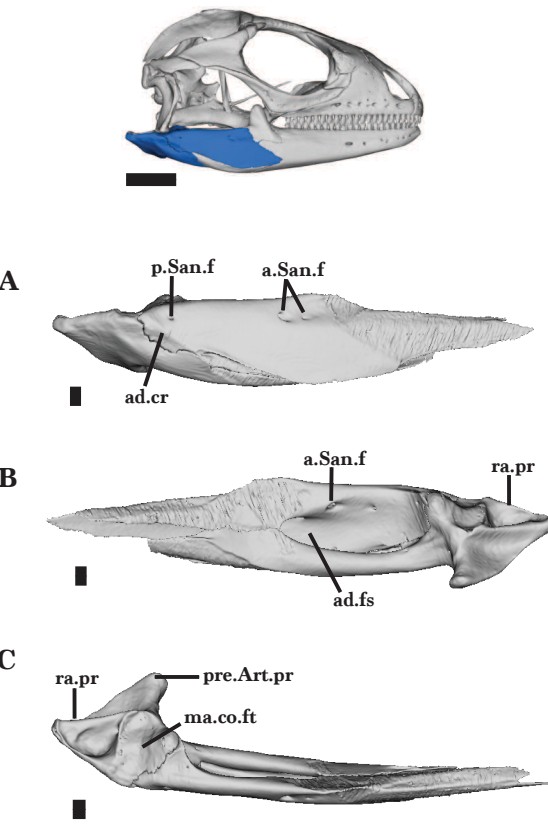

**Figure 3** **Right articular and surangular.** Right articular and surangular of *Cyclura carinata* UF:Herp:32820 (ark:/87602/m4/M59620). (A) Lateral view, (B) medial view, (C) ventral view. Scale bar = 1 mm, 10 mm for reference skull.

### Angular

The angular is an elongated bone that curves along its ventral margin, forming much of the ventral border of the mandible. It is characterized by anterior and posterior processes that both taper into sharp ends. The angular contacts the fused articular and surangular bone posterodorsally, the dentary ventrally, and the splenial anterodorsally (Figs. 4A and 4B). Anteriorly, the angular underlies the posteroventral process of the splenial and contributes to the posterior region of the Meckelian fossa. In ventral view, the angular anteriorly separates into two long processes that create a v-shaped facet for the angular process of the dentary. The posterior process underlies the articular and slightly overlaps the posterolateral face of the surangular. The angular is pierced medially by the posterior mylohyoid foramen (Fig. 4A).

### Articular and surangular

The articular and surangular are fused elements that form the posterior portion of the mandible (Fig. 4). The suture of the two bones is visible medially in the segmented model, but the bones are indistinguishably fused posteriorly in the adult specimen (Figs. 3A and 3B) and are also not distinguishable in the juvenile specimen, though scan quality

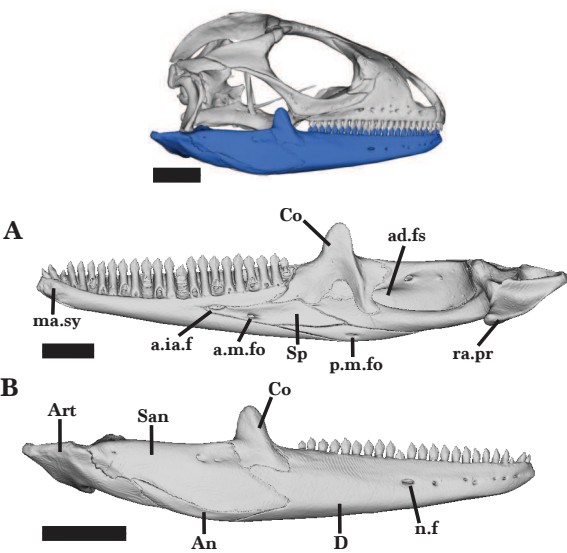

**Figure 4 Right Mandible overview.** Right mandible of *Cyclura carinata* UF:Herp:32820 (ark:/87602/m4/M59620). (A) Medial view, (B) lateral view. Scale bar = 6 mm for 4A, 10 mm for reference skull and 4B.

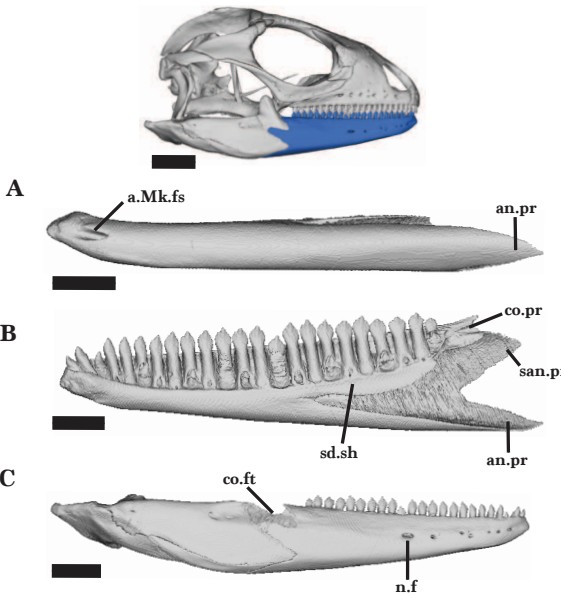

**Figure 5 Right dentary.** Right dentary of *Cyclura carinata* UF:Herp:32820 (ark:/87602/m4/M59620). (A) Ventral view, (B) medial view, (C) lateral view. Scale bar = 5 mm, 10 mm for reference skull.

may contribute for that specimen. Anteriorly, the broad dentary process extends past the anterolateral process of the coronoid (Figs. 4A and 4B). Together, the fused articular and surangular contact the angular ventrally, the coronoid anterodorsally and medially, the

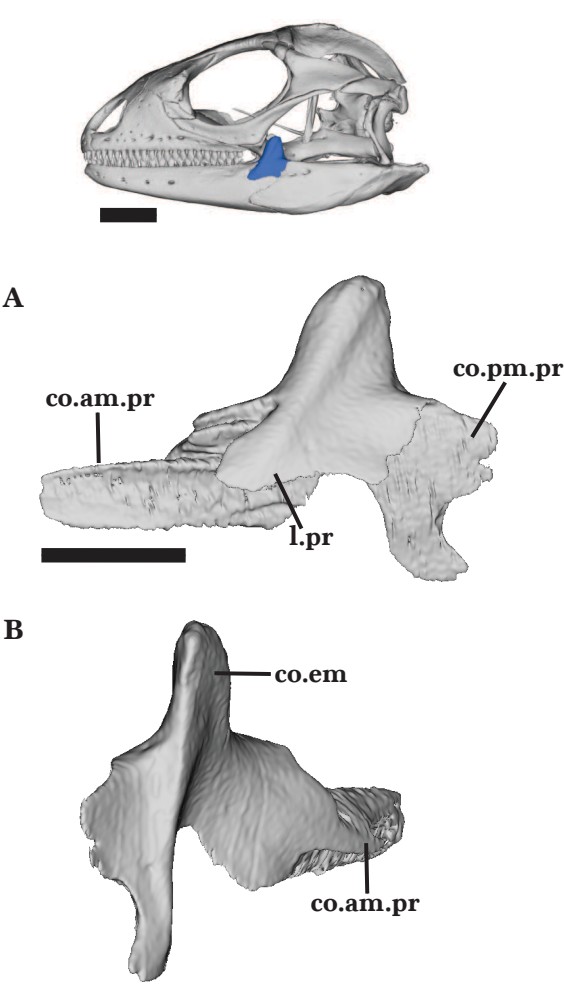

**Figure 6** **Left coronoid.** Left coronoid of *Cyclura carinata* UF:Herp:32820 (ark:/87602/m4/M59620). (A) Lateral view, (B) posteromedial view. Scale bar = 5 mm, 10 mm for reference skull.

dental anteriorly, the splenial anteromedially, and the quadrate posterodorsally (Figs. 2 and 3A).

On the medially curved surangular, there is a large facet for articulation with the medial processes of the coronoid and the anterior process of the articular (Fig. 3B). The surangular tapers anteriorly into a bifurcated end that contributes to the posterolateral border of the Meckelian fossa. A deep adductor fossa is bounded by the surangular and articular, and opens posteriorly (Fig. 3B). From the posterior end of the surangular, a prominent lateral adductor crest extends anteroventrally onto the angular (Fig. 3A). The articular and surangular are completely fused just posteroventral to that crest.

In medial view, the articular tapers from the large retroarticular process to the thin anterior process. The articular also extends anteriorly to a bifurcated end that inserts into the Meckelian fossa, contacting the splenial medially and the angular both ventrally and laterally (Fig. 4A). A large depression on the dorsal surface of the retroarticular process

meets the posterior end of the surangular (Fig. 3B). The retroarticular process expands into a posterior knob-shaped projection and a rounded, medially projecting prearticular process (angular process of *Oelrich, 1956*). There is a small facet that receives the mandibular condyle of the quadrate dorsally (Fig. 3C). Just posterior to the condyle facet, a foramen pierces the dorsal surface of the retroarticular process for the chorda tympani branch of the facial nerve (*Porter & Witmer, 2015*). Both the anterior and posterior surangular foramina are located on the lateral face, but only the anterior foramen pierces the surangular to form a large medial opening (Figs. 3A and 3B). A wide septum divides the anterior foramen. The septum is more noticeable on the left surangular compared to the right surangular.

## Marginal dentition

*Cyclura carinata* possesses pleurodont dentition. The adult *Cyclura carinata* specimen UF Herp 32820 bears three teeth on each side of the premaxilla with a single, small medial tooth in the process of being replaced (Fig. 7A). The premaxillary teeth are conical and unicuspid with slight shoulders above the pointed tip. The juvenile UMMZ 117401 also possesses seven unicuspid teeth, whereas the adult or subadult specimen MVZ Herp 81381 has eight teeth. Both maxillae of UF Herp 32820 bear 20 teeth and all tooth positions are filled. The first two anterior teeth of the left maxilla are unicuspid (Figs. 8A and 8B). Maxillary teeth 3–9 are generally uniform in size and slightly shouldered, while teeth 10–20 decrease in size distally and are tricuspid or multicuspid, though smaller accessory cusps beyond the first three are difficult to definitively distinguish in the segmented model (Fig. 8A) or scan volume renderings. The juvenile has 14 teeth on both the right and left maxillae. Maxillary tooth positions 11–14 of the juvenile are tricuspid, but do not have prominent accessory cusps compared to the adult. Each dentary has 21 teeth with visible, deep resorption pits at the tooth bases for replacement that are located directly medial to the teeth (Figs. 5B and 5C). The first four anterior teeth of the dentary are unicuspid and conical and project anterodorsally, while teeth 5–21 are multicuspid (at least 3 and up to 5 clearly distinguishable cusps in both segmented models and volume renderings) with a taller, well-developed middle cusp (Fig. 5B). The dentary teeth increase in size from tooth position 1–14 and decrease from position 15–21. The juvenile has 15 teeth on both the right and left dentary.

## Premaxilla

The premaxilla forms the anterior wall of the skull and curves posterolaterally and posterodorsally. The body of the premaxilla is widest anteriorly at the midpoint of the alveolar plate, from which the premaxillary teeth protrude ventrally (Fig. 7A). The long nasal process contributes to the anterodorsal surface of the snout and narrows posteriorly to a point as the lateral sides bear facets to contact the nasal anteromedial processes (Fig. 1A). The lateral facet of the maxillary process extends posterolaterally to articulate with the anterior process of the maxilla. The anterior portion of the premaxilla protrudes laterally and anteriorly past the lower jaw. Anterior premaxillary foramina are visible dorsally along the anterior surface, four of which pierce through to connect with the medial ethmoidal foramen (Figs. 7A and 7C).

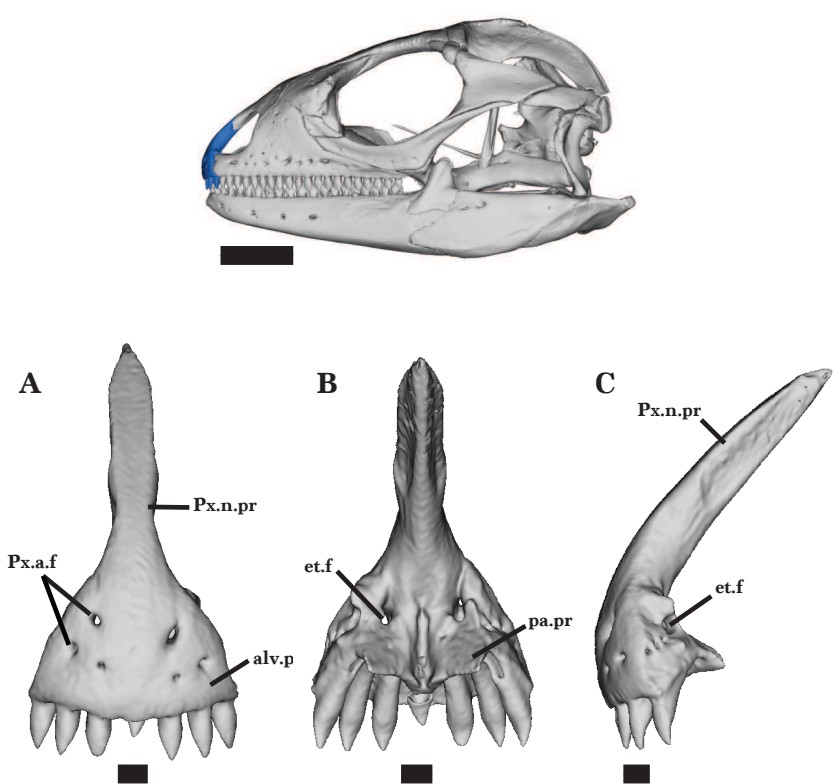

**Figure 7 Premaxilla.** Premaxilla of *Cyclura carinata* UF:Herp:32820 (ark:/87602/m4/M59620). (A) Anterior view, (B) posterior view, (C) lateral view. Scale bar = 1 mm, 10 mm for reference skull.

The external nares are partly formed by the posterior margins of the premaxilla but differ in shape among examined specimens. *Cyclura carinata* UF Herp 32820 has wide external nares that do not extend posteriorly past the nasal process of the premaxilla (Figs. 1A and 2). The external nares of *Cyclura carinata* MVZ Herp 81381 are narrow and more ovular in shape. The nasal processes of the adult specimens have a lateral flare at the midpoint of the process that the juvenile *Cyclura carinata* UMMZ 117401 lacks. The juvenile also lacks observable anterior premaxillary foramina, which are present in the adult or near adult specimens.

## Maxilla

The maxilla is a large, triradiate bone that forms the posterior portion of the upper jaw (Fig. 2). Both maxillae hold 20 teeth. The maxilla consists of a short premaxillary process (anterior process), a long jugal process (posterior process), and a large facial process (dorsal process). The rounded maxillary shelf flattens medially along the premaxillary process, which is short and blunt in lateral view. The premaxillary process curves medially, widening into anteromedial and anterolateral processes (Fig. 8C). The bifurcated end sits on top of the medial and lateral facets of the palatal process. The anteromedial process inserts between the palatal process of the premaxilla and the anterolateral face of the vomer. Although the vomer and maxilla do not contact each other, there are articulation facets

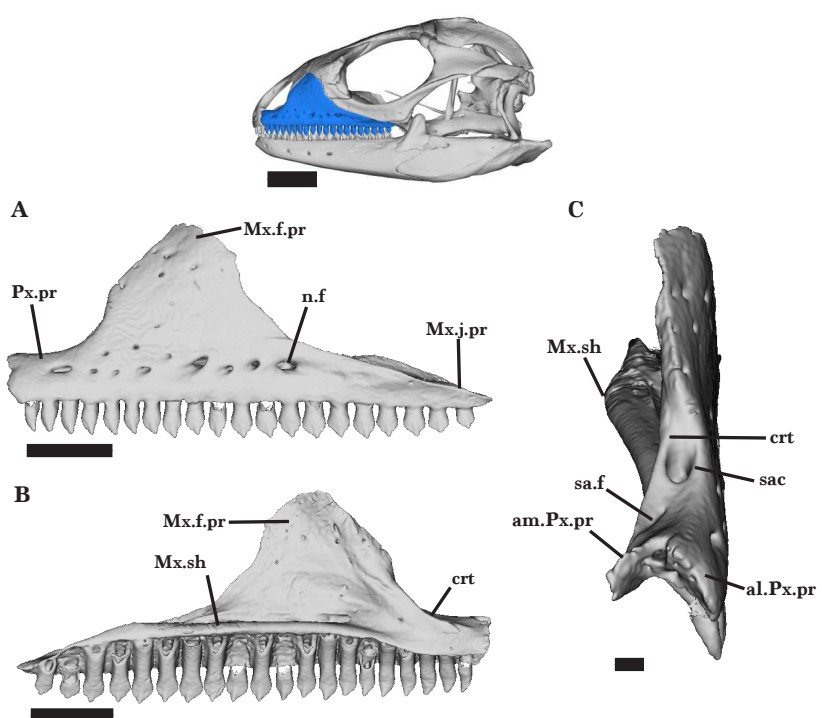

**Figure 8** **Left maxilla.** Left maxilla of *Cyclura carinata* UF:Herp:32820 (ark:/87602/m4/M59620). (A) Lateral view, (B) medial view, (C) anterior view. Scale bar = 5 mm, 10 mm for reference skull.

that suggest the presence of soft tissue connecting the two bones. The tall facial process curves dorsomedially to contribute to the anterior skull roof. The process is located slightly anterior to the midpoint of the maxilla (Fig. 8A). Dorsally, the facial process narrows to a V-shaped end that inserts between the nasal and prefrontal bones. The anterior edge of the facial process contacts the lateral process of the nasal, whereas the posterior edge contacts the anteroventral process of the prefrontal (Fig. 2). Just ventral to the prefrontal facet, a shallow lacrimal notch receives the anterior end of the lacrimal. The jugal process measures about one-third of the maxilla and carries many of the maxillary teeth (8 to 10 teeth). The jugal process underlies the anterior process of the jugal along its length, tapering posterodorsally to a sharp point (Fig. 8A).

The superior alveolar canal runs along the maxillary shelf, deepening anteriorly under the facial process (Fig. 8B). The large dorsal opening of the canal is visible just anterior to the facial process and is almost as wide as the premaxillary process itself (Fig. 8C). The small subnarial arterial foramen is located medial to the crista transversalis and just anterior to the dorsal opening of the superior alveolar canal (Fig. 8C). Large nutrient foramina are aligned along the lateral face of the maxilla and pierce into the superior alveolar canal. The right maxilla has seven nutrient foramina and the left maxilla has nine. Additional small foramina are located irregularly on the lateral surface of the maxilla, only a few of which pierce the facial process (Fig. 8A).

The dorsal opening of the superior alveolar canal is present in all of the adult specimens, as well as the small subnarial arterial foramen. *Cyclura carinata* MVZ Herp 81381 lacks both openings on the right maxilla, and the juvenile *Cyclura carinata* lacks both subnarial foramina. The maxillary teeth of the juvenile *Cyclura carinata* specimen are not well-developed and only teeth 11–14 are tricuspid.

## Nasal

The nasal is a paired bone that contributes to the anterodorsal margin of the skull. The nasals meet medially along the midbody of the two bones, diverging both anteriorly and posteriorly. Anteriorly, the nasals diverge laterally to form two separate processes: an anteromedial process that meets the premaxilla medially, and an anterolateral process (supranarial process of *Gauthier et al., 2012*) that meets the maxilla ventrolaterally (Figs. 1 and 2). The anterior processes exhibit a greater degree of divergence because the distance between the anterior processes is much larger than that of the posterior processes. The anterior processes narrow anteriorly and the posterior processes narrow posteriorly; however, the tips of the anterior processes taper into narrower, sharper points than do the posterior processes (Fig. 9A).

The posterodorsal face of the nasals is relatively flat, sloping anteroventrally along the anterior processes. The anteromedial processes are medially bounded by the lateral walls of the premaxilla nasal process. The anterolateral process slopes ventrally to articulate with the anterodorsal surface of the maxillary facial process and contacts the prefrontal (Fig. 9A).

At midbody, the nasal articulates with the prefrontal posteriorly and borders the anteromedial margin of the prefrontal laterally. The posterior process flattens posteriorly and rests on top of the anterodorsal facets on the frontal.

A short, sharp anteromedial process of the frontal slots into the space between the posteromedial borders of the nasals formed by the divergence of the two nasals (Fig. 9A). The distribution of foramina visible on the dorsal surface is mostly random, except for the three lateral foramina located in a line along the anterolateral process (Figs. 9A and 9B). The nasals of the juvenile *Cyclura carinata* UMMZ 117401 diverge posteriorly and there is a large medial gap along the posterior half of the nasals.

## Frontal

The frontal is a long bone that contacts the nasals anteriorly and widens posteriorly to meet the parietal, contributing to the dorsal border of the skull and orbit (Fig. 1A). The anterior region of the frontal bone widens slightly and is composed of two lateral and one medial process that each narrows to a point. The space between the anterior processes forms facets on which the nasal posterior processes sit (Fig. 10). The anteromedial process is short and sharp, whereas the anterolateral processes are slightly rounder at the tips. Additionally, the anterolateral processes slightly flare at mid-length before ultimately narrowing to a tip (Fig. 10). The posterior processes narrow into much sharper tips than the anterolateral processes. The posterior region of the frontal diverges laterally into two posterolateral processes as it widens to about twice the width of the anterior region. The posterior

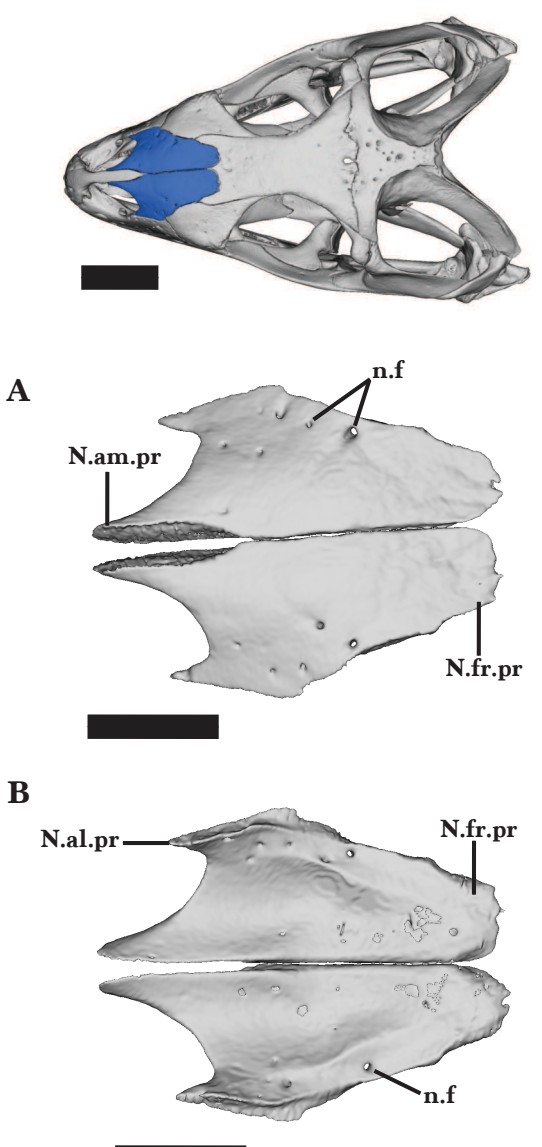

**Figure 9** **Nasals.** Nasals of *Cyclura carinata* UF:Herp:32820 (ark:/87602/m4/M59620). (A) Dorsal view, (B) ventral view. Scale bar = 5 mm, 10 mm for reference skull.

processes elongate laterally to overlap the medial process of postfrontal, extending past the lateral bounds of the anterolateral processes (Figs. 10A and 10B).

The frontal meets the anterior edge of the parietal along the entirety of the posterior margin (Fig. 10A). There is a posteromedial protrusion that extends posteriorly as two short, separate projections that converge medially, enclosing the parietal foramen (Figs. 10A and 10B). The posteromedial protrusion contributes to the triradiate anteromedial border of the parietal, resulting in an indent on each side of the protrusion. In both adult and juvenile *Cyclura carinata* specimens, the dorsal surface is slightly convex.

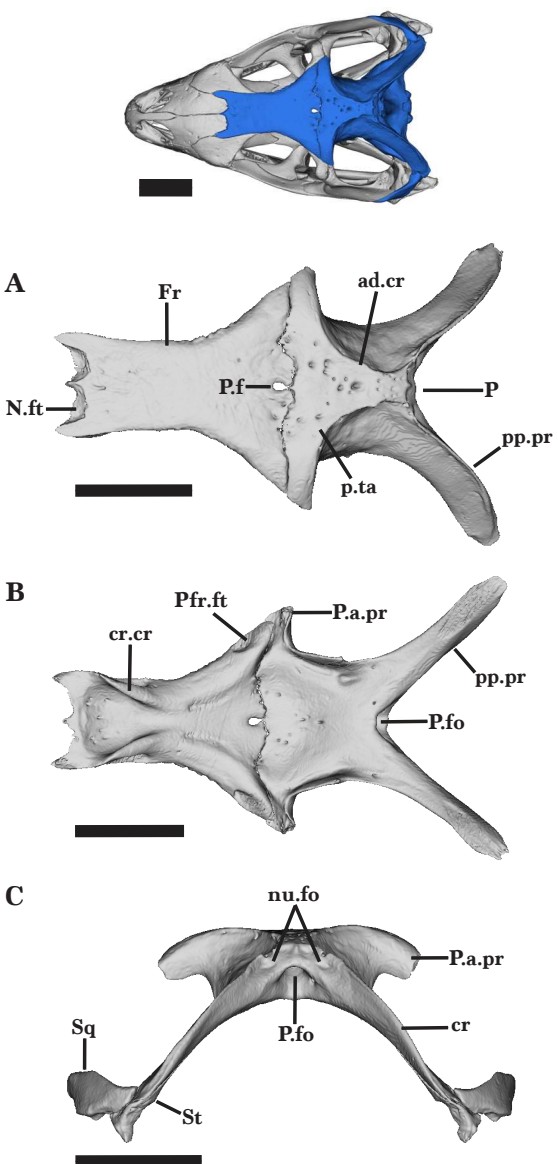

**Figure 10 Frontal and parietal.** Frontal and Parietal of *Cyclura carinata* UF:Herp:32820 (ark:/87602/m4/M59620). (A) Dorsal view, (B) ventral view. Scale bar = 10 mm.

## Parietal

The parietal is composed of two anterior and two posterior processes that extend distally from a rectangular, dorsal midbody. Located posteriorly on the skull, the parietal contributes to the posterodorsal roof of the skull and dorsal border of the upper temporal fenestra (Fig. 1A). The ventral surface of the parietal is concave.

The midbody of the parietal slopes slightly anteriorly, creating a shallow dorsal depression from which two anterior processes diverge laterally to form the anterior border of the parietal. The parietal table widens anteriorly, eventually diverging laterally to form the

two anterior processes that articulate with the posterior edge of the frontal (Fig. 10A). The posterior half of the parietal table experiences the inverse, as it narrows to meet the anterior points of the adductor crests. The descending processes located below the dorsal shelf broaden ventrolaterally and demarcate the lateral bounds of the midbody (Figs. 10A and 10B). Nonsymmetrical foramina occupy the dorsal face of the parietal, varying in size as they spread anterolaterally (Fig. 10A).

The anterior processes extend laterally while the tips curve slightly posterolaterally. The blunt, flat ends of the processes reach slightly past the posterior processes of the frontal, articulating laterally with the posterior process of the postfrontal and the dorsomedial process of the postorbital (Fig. 10A). Furthermore, the anterior processes exhibit a minor ventrolateral slope, contributing to the anterodorsal bounds of the temporal region. At the frontoparietal contact, a triradiate anteromedial structure gives rise to two larger, more rounded lateral protrusions and a short, sharp medial protrusion that inserts into the posterior opening of the parietal foramen (Fig. 10A).

The posterior region of the parietal contributes dorsally to the temporal and posterior regions of the skull and is mainly characterized by two long postparietal (supratemporal) processes. The anterior processes are wider than the posterior processes but are much shorter. The postparietal processes extend further laterally than the anterior processes (Figs. 10A and 10B). Elongating distally from the dorsal midbody, the processes curve posterolaterally and taper in width along the entire length. Along this curve, the processes twist ventromedially such that the lateral surface faces slightly dorsally (Fig. 10C). The postparietal processes are pinched dorsally to form the supratemporal crest that runs along the processes, flattening just short of the straight ends of the processes. The distal end of the process meets the supratemporal.

The posterodorsal surface of the parietal contains two nuchal fossae located at the base of each postparietal process, medial to the dorsal crest of the postparietal processes (Fig. 10C). The bilateral nuchal fossae are weakly developed and present as small, shallow depressions. There is a large parietal fossa located medially on the posterior wall, just ventral to the nuchal fossa. The parietal fossa deepens anteriorly into the midbody of the parietal.

All observed specimens of *Cyclura carinata* have a fairly smooth dorsal surface and lack a parietal crest (Fig. 10A). On the dorsal surface, the parietal foramen is present in *Cyclura carinata* UF Herp 32820, but not discrete in the juvenile. In *Cyclura carinata* UF Herp 32820 and MVZ Herp 81381, the parietal foramen is ovular and narrow; there is a thin posterior opening that receives an anterior projection of the parietal (Figs. 10A and 10B). The juvenile *Cyclura carinata* UMMZ 117401 does not have this small foramen, but instead exhibits a large frontoparietal fontenelle between the frontal and parietal that is almost as wide as the skull (*Hernández-Jaimes, Jerez & Ramírez-Pinilla, 2012*). The dorsal surface of the juvenile *Cyclura carinata* is smooth and does not have any additional foramina. The parietal of the juvenile is mediolaterally constricted and does not contact the processus ascendus of the supraoccipital.

### Prefrontal

The prefrontal is a robust, triradiate bone that has a posteriorly extending orbital process (posterodorsal process), an anteromedial process, and a posteroventral process. Together, the processes contribute to the anterior margin of the orbit as well as to the anterior portion of the skull roof (Fig. 2). The prefrontal articulates with the frontal posteromedially, the nasal anteromedially, and the palatine ventrally. Anteriorly, the ventral portion of the anteromedial process underlies the lacrimal and maxillary facial processes and the dorsal portion underlies the lateral process of the nasal (Fig. 11B). The prefrontal does not contact the jugal.

Posteriorly, the orbital process projects posterodorsally, tapering to a sharp point (Fig. 11A). The orbital process is relatively flat and much longer than the other two processes. The process articulates with the frontal medially, extending along the anterolateral process of the frontal. The posteroventral process does not extend posteriorly past the end of the orbital process. The posteroventral process curves ventrolaterally to articulate with the medial shelf of the lacrimal and enclose the lacrimal foramen (Fig. 11A). The end of the posteroventral process fits into a notch on the palatine (Fig. 11C). Dorsally, a prominent ridge runs along the process to form a deep medial concavity. A shallow prefrontal foramen is visible on the dorsal ridge of the posteroventral process, as well as additional small foramina on the dorsal and medial surfaces. The prefrontal of the juvenile has a prominent posteroventral process, similar to that of the adult. However, the anteromedial process of the juvenile is not well-developed and does not fully articulate the lateral margins of the nasal. Posteriorly, there is a small gap in between the two bones. The juvenile does not bear additional foramina on the dorsal surface.

### Lacrimal

The lacrimal is a small bone that contributes to the anteroventral border of the orbit. The anterior margin of the lacrimal has a slight dorsolateral curve, slotting into the lacrimal notch of the maxillary facial process (Fig. 12A). The lacrimal exhibits a shallow groove along the lateral face. The anterodorsal edge of the lacrimal is flat and approximately half the width of the ventral margin. There is a relatively long posteroventral process that extends posteriorly to reach the entirety of the dorsal edge of the jugal anterior process (Fig. 11C).

The short medial shelf of the lacrimal articulates with the anterolateral process of the palatine and the posteroventral process of the prefrontal to fully enclose the small, oval-shaped lacrimal foramen (Fig. 11A). The medial shelf forms the ventral border of the foramen while the prefrontal process forms the dorsal border.

### Jugal

The jugal anteriorly contributes to the ventral margin of the orbit and posteriorly contributes to the temporal region of the skull. Located laterally on the skull, the jugal contacts the maxilla anteroventrally, the lacrimal anterodorsally, the ectopterygoid ventromedially, and the postorbital posterodorsally (Figs. 12A and 12B). The long posterior (postorbital) process flares laterally to the medial process of the jugal, from which the anterior process curves medially to meet the lacrimal.

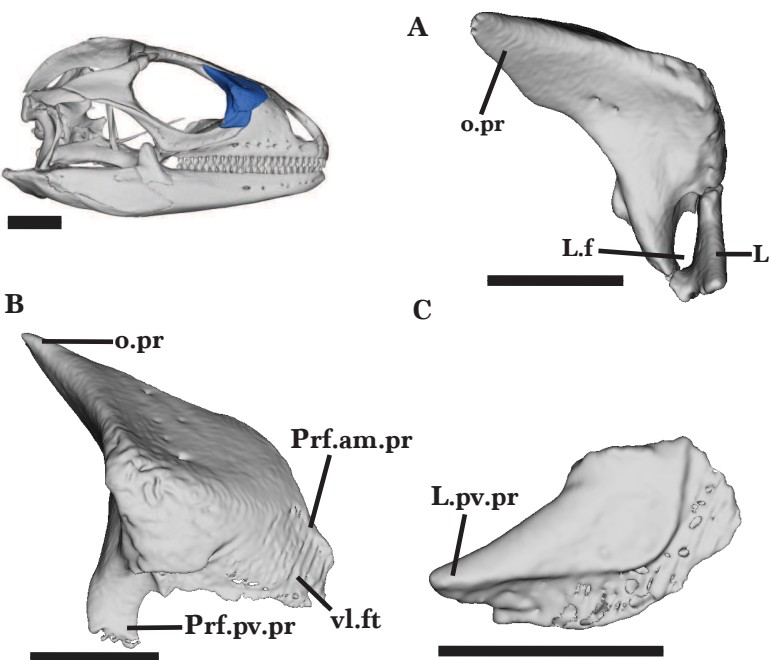

**Figure 11 Right Prefrontal and Lacrimal.** Lacrimal and prefrontal of *Cyclura carinata* UF:Herp:32820 (ark:/87602/m4/M59620). (A) Right prefrontal and lacrimal in posterior view, (B) right prefrontal in lateral view, (C) right lacrimal in lateral view. Scale bar = 5 mm, 10 mm for reference skull.

In lateral view, the jugal is the widest at the midpoint, tapering anteriorly to a flat end and posteriorly to a sharp point (Fig. 12A). The ventromedial process contacts the ectopterygoid medially and is characterized by a small protrusion that angles slightly posteriorly (Fig. 12B). From the ventromedial process, both the anterior and posterior processes have a shallow groove running along the medial faces. Small foramina run along the lateral face of the anterior process, nearing the ventral margin. The right jugal bears more lateral foramina than the left jugal.

The anterior orbital (maxillary) process is approximately the same length as the postorbital process, but is much wider (Fig. 12A). Extending anterodorsally, the anterior process has a flat, wide end that articulates with the ventral margin of the lacrimal. The entire anterior process ventrally articulates with the posterodorsal margin of the maxilla.

The posterior (postorbital) process extends posterodorsally to articulate with the anteroventral face of the postorbital. The postorbital process tapers as it posteriorly elongates, and is approximately one-third the width of the anterior process (Fig. 12C). The jugal of the adult and juvenile specimens are similar in shape and articulation with surrounding bones, but the juvenile jugal does not bear foramina along the lateral face.

## Ectopterygoid

The ectopterygoid is a short, thick bone that extends anterolaterally to contribute to the posterolateral border of the suborbital fenestra (Figs. 13A and 13B). The bone is constricted at the midpoint in dorsal view and has an anterior jugal process and a posterior pterygoid

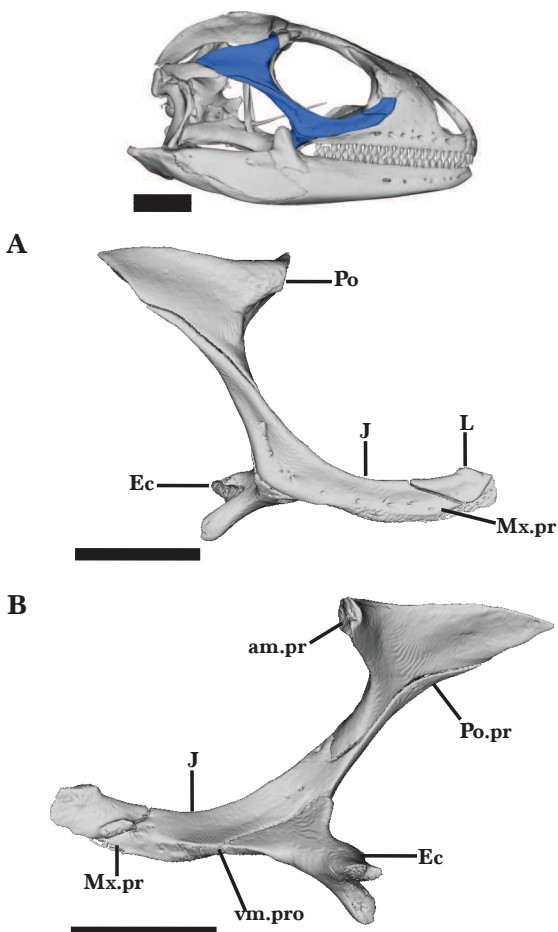

**Figure 12 Right lacrimal jugal postorbital ectopterygoid.** Right lacrimal, jugal, ectopterygoid, and postorbital of *Cyclura carinata* UF:Herp:32820 (ark:/87602/m4/M59620). (A) Lateral view, (B) medial view. Scale bar = 10 mm.

process. The jugal process is thickest posteriorly, where the lateral ectopterygoid spur (lateral exposure of the ventral corner of the ectopterygoid or wedge structure of *Smith, 2009*) articulates with the jugal, and tapers anteriorly along the ventromedial surface of the jugal (Fig. 12C). The lateral spur has a prominent lateral exposure ventral to the pterygoid, and a portion of the structure inserts between the jugal ventral process and the maxilla orbital process. Posteriorly, the pterygoid process is enlarged and bifurcated to clasp the ventrolateral flange of the pterygoid (Figs. 12B and 12A).

## Postfrontal

The postfrontals of *Cyclura carinata*, including the juvenile specimen, are short and project laterally along the anterior margin of the postorbital. The postfrontal is a small, crescent-shaped bone that forms the posterodorsal margin of the orbit (Fig. 1). The anteromedial process is elongated and sharper relative to the rounded posterolateral process (Fig. 2). The width of the anteromedial process tapers anteriorly, and it medially borders the

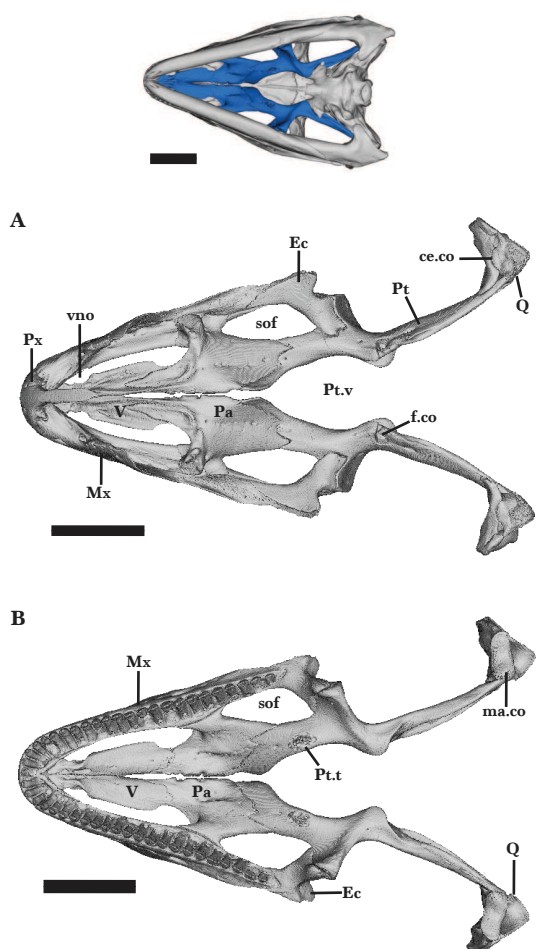

**Figure 13 Full palatal bones.** Palatal bones of *Cyclura carinata* UF:Herp:32820 (ark:/87602/m4/M59620). (A) Dorsal view, (B) ventral view. Scale bar = 10 mm, 10 mm for reference skull.

posterolateral corner of the frontal. The posterolateral process is a bulbous structure that articulates with the anterodorsal face of the postorbital (*Avery, 1970*). The posteromedial process contributes to the upper temporal fenestra and posteriorly contacts the parietal and postorbital. Medially, the anterior process of the postfrontal underlies the frontal and parietal bones.

## Postorbital

The postorbital is a long, triradiate bone that forms the posterior margin of the orbit. Dorsally, the anterior process articulates anteriorly with the postfrontal and anterolaterally with the parietal (Fig. 2). The postorbital narrows posteriorly to meet the squamosal posteroventrally. The ventral face of the anteroventral process articulates with the jugal, extending anteriorly as it overlaps the posterior process of the jugal dorsomedially (Figs. 12A and 12B). This jugal process is longer than the anterodorsal and posterior processes in both the adult and juvenile.

### Squamosal

The squamosal is a small, slender bone that slopes anteroventrally. Located in the temporal region, the squamosal lies in between the postorbital, quadrate, and supratemporal bones (Fig. 2). The anterior process underlies the posteroventral surface of the postorbital without contact, leaving a small gap remaining between the two bones that suggests the presence of soft tissue articulation. The posteroventral face articulates with the posterodorsal surface of the quadrate, while the small posteromedial (dorsal) process laterally articulates with the posterolateral face of the supratemporal. The posteroventral process slots into the squamosal notch on the dorsal surface of the quadrate.

### Supratemporal

The supratemporal is a long, slender bone located in the posterior region of the skull that contacts the quadrate, squamosal, parietal, and braincase (*i.e.,* the paroccipital process). The posterior process is rounded and lies in between the squamosal, postparietal process, paroccipital process, and cephalic condyle (Fig. 14C). The supratemporal extends anteriorly along the medioventral face of the postparietal process, tapering to a sharp point. The rounded posterior process has a small notch that accommodates the posterior process of the squamosal. This notch creates a shallow groove that flattens along the anterior process. The medial face of the posterior process articulates with the paraoccipital process (Fig. 10C). The thin dorsal surface fits in between the squamosal and paroccipital process, preventing contact between the two. The posterior process lies on top of the cephalic condyle to contact the quadrate ventrally. The supratemporal of the juvenile is not well-developed and is indistinguishable in articulation with the postparietal process of the parietal.

### Quadrate

The quadrate is a robust, vertically oriented bone located in the posterior of the skull. The bone contacts the supratemporal and squamosal dorsally, the pterygoid ventromedially, and the articular ventrally (Fig. 2). The quadrate is narrow ventrally but widens dorsally to form the broad, well-developed cephalic condyle. The cephalic condyle is slightly sloped posteroventrally and is overlapped by the squamosal anteriorly and the supratemporal dorsomedially (Fig. 14D). The small ventromedial knob of the squamosal fits into the squamosal notch located on the dorsal surface of the quadrate.

The column of the quadrate extends along the vertical length of the bone and is laterally bounded by a deep quadrate conch (Fig. 14C). A wide flange extends medially from the column. This flange is wider and has a depression deeper than the conch. The quadrate bears a lateral tympanic crest that thickens slightly anterior to the conch, and reaches the cephalic condyle dorsally (Fig. 14B). A small, shallow pterygoid lamina is present on the medial flange of the quadrate above the mandibular condyle (Fig. 14C). It articulates medially with the quadrate process of the pterygoid. The mandibular condyle sits on top of the articular and is characterized by round lateral and medial condyles.

The ossification on top of the cephalic condyle forms a pitted structure. Separated by the central column, two additional foramina are present above the mandibular condyle. Both foramina pierce the quadrate and are visible on the anterior and posterior faces of

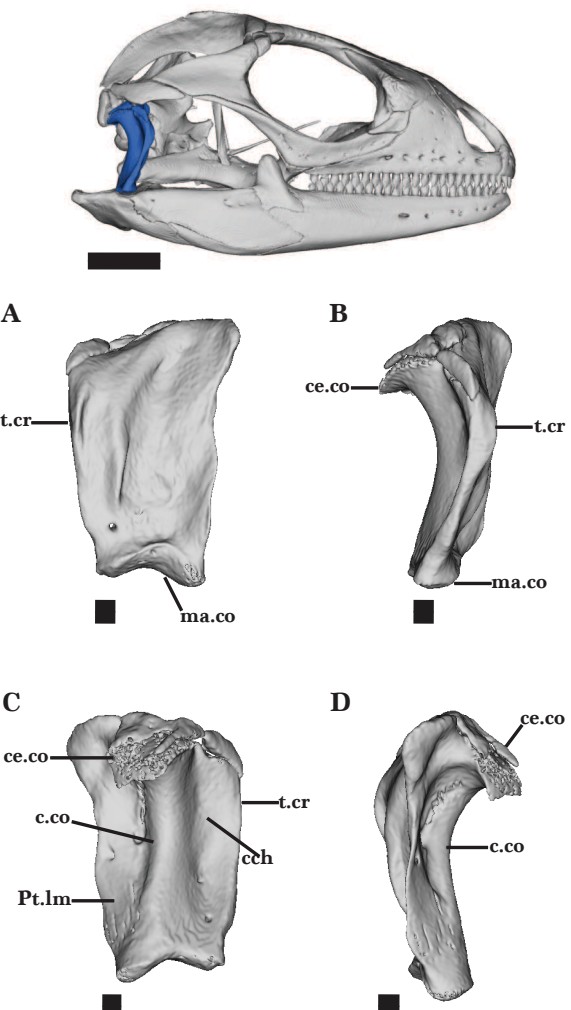

**Figure 14 Right quadrate.** Right quadrate of *Cyclura carinata* UF:Herp:32820 (ark:/87602/m4/M59620). (A) Anterior view, (B) right lateral view, (C) posterior view, (D) left lateral view. Scale bar = 1 mm, 10 mm for reference skull.

the quadrate (Figs. 14A and 14C). The medial foramen, however, is much larger than the lateral foramen on both faces.

Adult specimens of *Cyclura carinata* possess ossified cartilage on top of the cephalic condyle that contributes to the tight articulation of the quadrate, the squamosal, postparietal process, and the supratemporal (Figs. 14C and 14D). This calcified cartilage is not present on the quadrate of the juvenile *Cyclura carinata*.

## Pterygoid

The pterygoid is a long, triradiate bone that is located in the posterior region of the palate (Figs. 13A and 13B). The interpterygoid vacuity separates the pterygoid from its contralateral element. The pterygoid contacts the epipterygoid dorsally, palatine and ectopterygoid anteriorly, and quadrate posterolaterally (Figs. 13A and 13B).

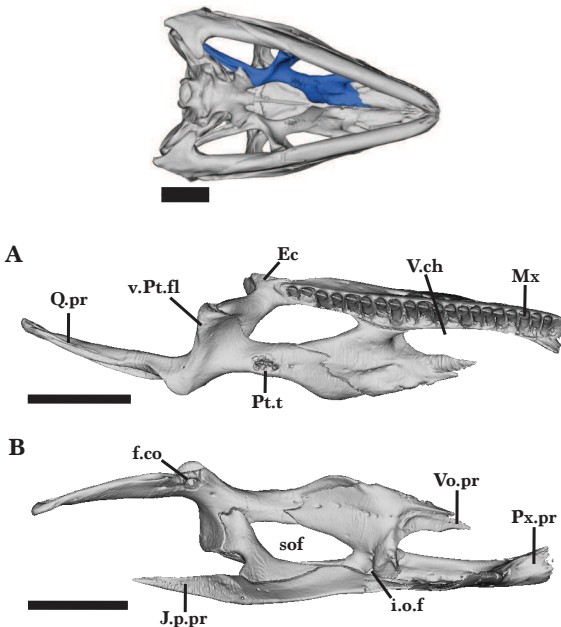

**Figure 15  Right palatal bones.** Right palatal bones of *Cyclura carinata* UF:Herp:32820 (ark:/87602/m4/M59620). (A) Ventral view, (B) dorsal view. Scale bar = 10 mm.

The anterior palatine process is relatively flat dorsoventrally and thickens medially. The anterolateral margin of the process is slightly concave and laterally contributes to the posteromedial border of the suborbital fenestra (Fig. 15A). In dorsal view, the palatine process expands anteriorly to slot into the dorsal pterygoid facet of the palatine. A small notch on the dorsal surface of the palatine process of the pterygoid receives the posterolateral projection of the palatine (Fig. 16A). A small anterior foramen is present on the shallow dorsomedial groove of the palatine process. The pterygoid expands ventrally from the midpoint of the bone into a short ventromedial flange and a larger ventrolateral flange (transverse process of *Oelrich, 1956*). The small ventromedial flange is rounded along its medial edge and has a thin dorsal groove that bears a foramen, which pierces the flange medially. The ventrolateral pterygoid flange is curved anterolaterally and has an expanded, notched end that forms a joint articulation with the posterior process of the ectopterygoid (Fig. 15A). An additional large foramen is visible on the dorsal surface, just anterior to the ectopterygoid facet. Dorsal to the ventromedial pterygoid flange, there is a small dorsal facet that receives the ventral end of the ectopterygoid. The fossa columella opens dorsally at the base of the posterior quadrate process of the pterygoid (Fig. 16B). The elongate quadrate process extends posterodorsally to articulate with the pterygoid lamina of the quadrate. At the base of the process, a short ridge forms on the dorsal surface just dorsal to the fossa columella.

Small pterygoid teeth are present on the ventromedial face of the palatine process, all of which are similar in size and conical shape (Figs. 15A and 16A). On the adult *Cyclura carinata* UF Herp 32820, the left pterygoid has six to eight pterygoid tooth positions and

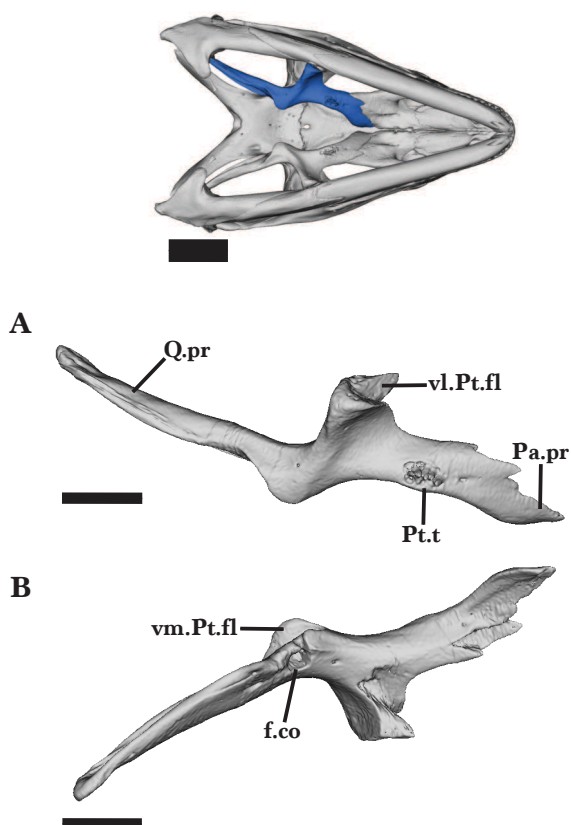

**Figure 16  Right pterygoid.** Right pterygoid of *Cyclura carinata* UF:Herp:32820 (ark:/87602/m4/M59620). (A) Anterior view, (B) dorsal view. Scale bar = 5 mm.

one empty position, whereas the right pterygoid has five teeth and two empty positions (Figs. 13B and 16A). The juvenile does not exhibit observable pterygoid teeth.

## Epipterygoid

The epipterygoid is a long, columnar bone that extends dorsally from the fossa columella of the pterygoid towards the epipterygoid process of the parietal. The epipterygoid is slightly inclined posterodorsally and is laterally convex at the midpoint. The dorsal end is mediolaterally compressed. A shallow groove widens along the ventrolateral surface of the epipterygoid and is anteroposteriorly compressed. The epipterygoid does not reach the descending (epipterygoid) process of the parietal, leaving a large dorsal gap between the two bones. The rounded, ventral end of the epipterygoid fits into the fossa columella of the pterygoid (Fig. 16B). In both the adult and juvenile specimens, the epipterygoid does not touch the descending process of the parietal.

## Septomaxilla

The septomaxilla is a thin paired bone located in the vomeronasal region, just posterior to the premaxilla and dorsal to the vomers. A small medial space that is widened anteriorly separates the septomaxillae. The septomaxilla is characterized by the tapering anterior

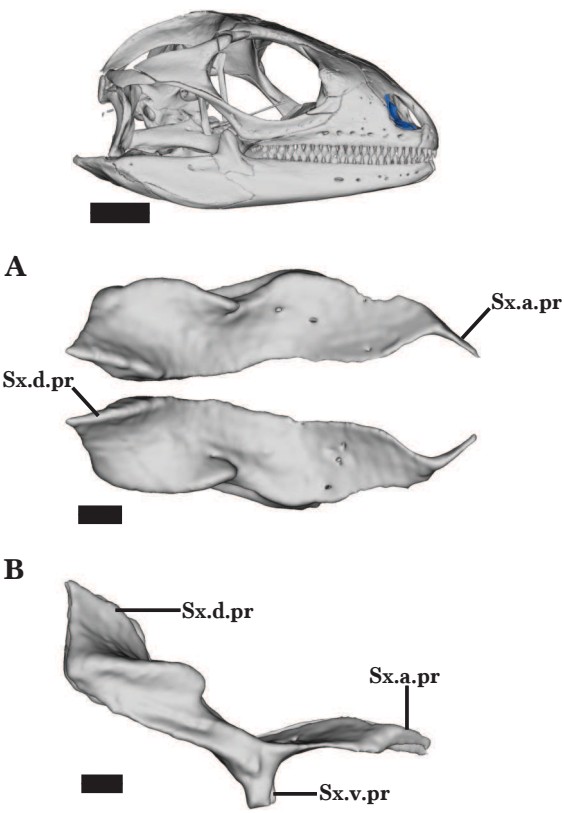

**Figure 17** **Septomaxillae.** Septomaxillae of *Cyclura carinata* UF:Herp:32820 (ark:/87602/m4/M59620). (A) Anterior view, (B) lateral view. Scale bar = 1 mm, 10 mm for reference skull.

process, broad dorsal process, and small ventral projection. In dorsal view, the bone is horizontally oriented, whereas the posterior half of the septomaxilla is steeply inclined toward the skull roof (Fig. 17A).

The anterior end tapers to a point and curves medially along the medial margin of the anterior process of the maxilla. The right septomaxilla has a more distinct articulation with the maxilla whereas the left maxilla has a small gap between the two bones, indicating individual variation and suggesting intraspecific variation. The septomaxilla is laterally constricted, forming a short dorsal ridge and a small lateral notch (Fig. 17A). Viewed anteriorly, the posterior process is both medially and laterally raised into tall walls that form a deep dorsal groove. This dorsally projecting process thickens posteriorly before ending in a short point. At the midpoint of the septomaxilla, a ventral process projects from the lateral margin to contact the dorsolateral surface of the vomer (Fig. 17B). A shallow groove widens posteriorly on the ventral surface of the posterior process. There are small, randomly distributed foramina on the dorsal surface of the septomaxilla.

## Vomers

The vomer is a short, paired bone that is located ventral and posterior to the septomaxilla and that forms the anterior portion of the palate. The vomers taper anteriorly to a point,

contacting each other medially along most of their length, and widen posteriorly to contact the palatines posterodorsally (Fig. 13A). A short medial process protrudes from the anterior end of the left vomer and fits into a facet on the right vomer. The small, deep vomeronasal region sits just posterior to the anteromedial process and lateral to a medial crest. The tall medial crest extends dorsally along the vomer, forming a dorsal groove that widens posteriorly and a large facet that clasps the vomerine process of the palatine (Fig. 13A). The medial surface is pierced by small foramina that are not visible when the vomers are in articulation. Large foramina that occupy the dorsal groove are visible along the ventral surface of the vomer, including the foramen for the medial palatine nerve. An additional medial foramen located in the vomeronasal region pierces the bone, opening on the ventromedial margin (Fig. 13A). A raised anterolateral shelf extends from the vomeronasal region, separating the nasal region from the choana. On the ventral surface, a sharp ridge curves posterolaterally from the anterior tip to join the lateral crest of the vomer (Fig. 13B).

## Palatine

The palatine is a large, paired bone located between the vomer and pterygoid (Figs. 13A and 13B). The palatine has a ventrally sloped pterygoid process, a vomerine process, and two laterally projecting maxillary processes.

Anteriorly, the vomerine process tapers to slot into a large facet on the posterior process of the vomer. A thin crest extends from the medial crest of the vomer, forming a deep dorsal groove along the vomerine process to the base of the maxillary processes (Fig. 15B). There are small protrusions along the posteromedial margin of the vomerine process that indicate articulation with the vomer. A deep choanal groove is present on the ventral surface of the palatine (Fig. 15A). Two short lateral processes articulate with the maxilla, prefrontal, lacrimal, and jugal. The medial shelf of the lacrimal and the posteroventral process of the prefrontal fit into a dorsal depression on the dorsolateral process of the palatine. The ventrolateral process articulates with a facet on the medial surface of the jugal and sits on the posterior end of the maxillary shelf. The posterior pterygoid process is dorsoventrally flattened and narrows posteriorly to a bifurcated end with short medial and lateral projections (Fig. 15B). The posterior end has a dorsal facet that accommodates the anterior process of the pterygoid and a lateral projection that fits into a dorsal notch on the pterygoid. The ventrolateral process and posterior pterygoid process contribute to the anterior and medial borders of the suborbital foramen (Figs. 13B and 15B).

The infraorbital foramen opens anteriorly through the ventrolateral maxillary process, just above the maxillary shelf (Fig. 15B). The foramen is bounded dorsally and ventrally by the short lateral processes of the frontal, and laterally by the maxilla. Aligned along a shallow, dorsal depression on the pterygoid process, multiple small foramina pierce the palatine and are visible ventrally. Additional foramina occupy the vomerine process along the anteromedial crest and open medially into the space between the palatines. A small foramen is present on the dorsal surface of the dorsolateral process, just ventral to the articulation of the prefrontal. There are no palatine teeth present on the ventral surface of the element (Fig. 15B).

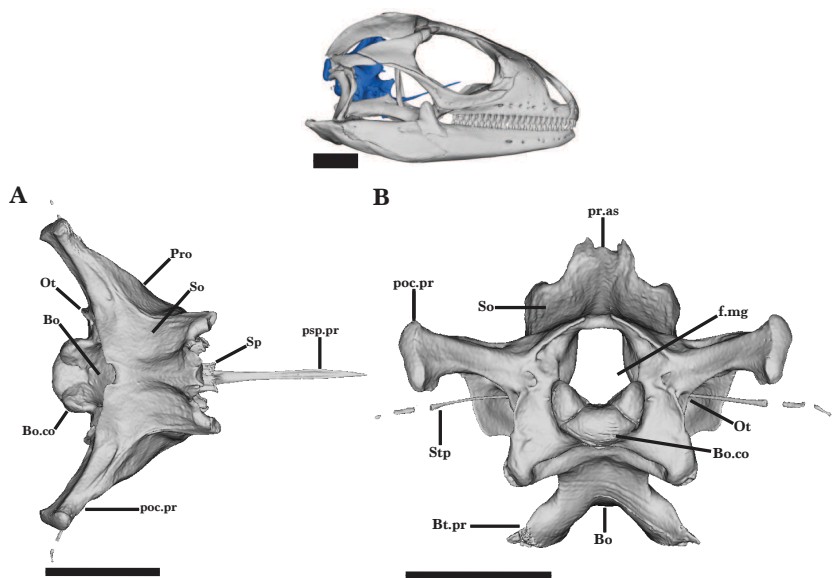

**Figure 18 General braincase 1.** General braincase of *Cyclura carinata* UF:Herp:32820 (ark:/87602/m4/M59620). (A) Dorsal view, (B) posterior view. Scale bar = 10 mm.

## Stapes

The stapes is a thin, cylindrical bone that slightly slopes ventrally as it extends posterolaterally from the braincase (Figs. 18B and 19B). Although both ends are expanded, the stapes is thickest at the round medial end. The medial end of the stapes is in the fenestra ovalis, whereas the lateral end extends into the space posterior to the quadrate (Fig. 19A). There are two isolated ossifications just posterior to each stapes. These fragments are present on both sides and extend along the same direction of the stapes, tapering to a small point.

## Braincase
### General features of the braincase

The braincase of *Cyclura carinata* is a posterior structure of the skull that articulates with the vertebral column *via* the occipital condyle (Fig. 18B). It is defined by five main elements: the anteroventral sphenoid, the posteroventral basioccipital, the dorsal supraoccipital, the paired lateral prootic, and the paired posterior otooccipital. Together, the elements possess articulations with the pterygoids, the stapes, the quadrate, the supratemporal, and the postparietal processes of the parietal (*Bever, Bell & Maisano, 2005*). The otooccipital is defined by a fusion of the exoccipital and the opisthotic elements. The fused otooccipital contains or contributes to large foramina, including the posterior foramen magnum and the fenestra ovalis (Fig. 18B). Sutures are not clearly distinguishable among any of the other braincase elements for UF Herp 32820, indicating complete fusion of the braincase. Most of the braincase is fused in MVZ Herp 81381 besides the prootic and the sphenoid, and UMMZ 117401 lacks braincase fusion.

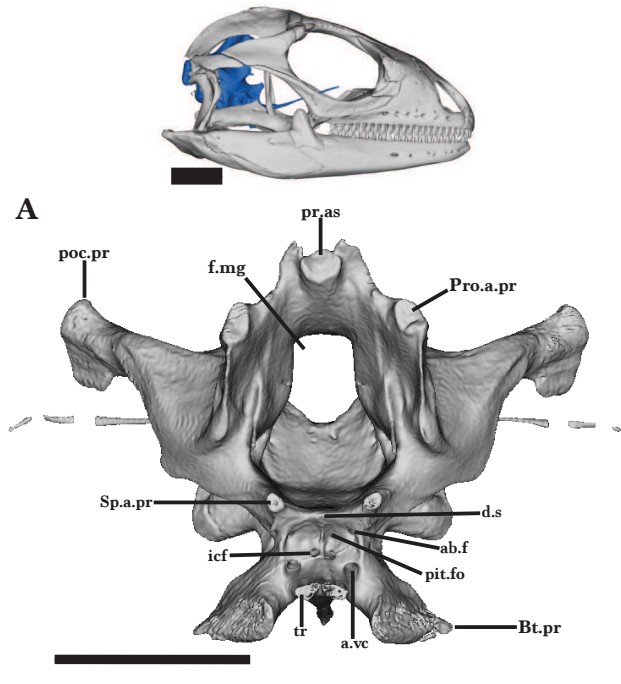

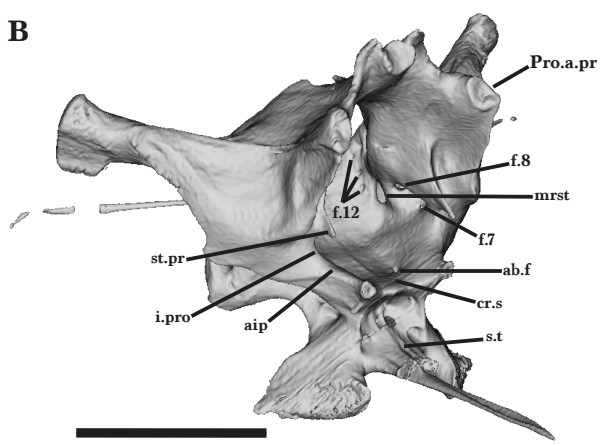

**Figure 19  General braincase 2.** General braincase of *Cyclura carinata* UF:Herp:32820 (ark:/87602/m4/M59620). (A) Anterior view, (B) anterolateral view. Scale bar = 10 mm.

## Orbitosphenoid

The orbitosphenoid is a paired, ossified element that is isolated from the other cranial bones. The orbitosphenoid forms from the ossification of the orbitotemporal cartilages pila metoptica and taenia medialis (*Tarazona & Ramírez-Pinilla, 2008*). Compressed dorsoventrally, the orbitosphenoid is located dorsal to the parasphenoid process. The orbitosphenoid extends anterodorsally towards the roof of the skull and the ends of the bone curve medially, forming a triradiate shape. The juvenile has a well-developed orbitosphenoid.

## Supraoccipital

The dorsal roofing of the braincase is formed by the supraoccipital. The supraoccipital also dorsally borders the foramen magnum. The supraoccipital meets the prootic ventrally and anterolaterally and meets the otooccipital posteriorly (Fig. 18A). The supraoccipital is concave in lateral view. Lateral flanges of the supraoccipital are triangular in shape and slope ventrally to slightly overlap the otooccipital.

The processus ascendens is short and broad in dorsal view and has two small lateral projections. The processus ascendens extends anteriorly to approach the deep parietal fossa but does not establish contact with the parietal. A weak midsagittal crest forms posteriorly on the dorsal surface of the processes ascendens (Fig. 18B). The dorsal surface is occupied by two wide, shallow depressions on both sides of the processes ascendens. Along the posterior edge, there is a prominent notch at the midline of the supraoccipital (Fig. 18A).

The lateral flange of the supraoccipital contributes to the dorsomedial roof of the cavum capsularis. The small groove of the utricular recess extends along the internal medial surface of the cavum capsularis, just dorsal to the osseous common crus. From the cavum capsularis, the large common crus deepens anteriorly to merge with the anterior and posterior semicircular canals (*Villa et al., 2018*). The anterior semicircular canal opens dorsally from the anterior ampullar recess towards the articulation of the supraoccipital and prootic, and the posterior semicircular canal opens posterolaterally towards the otooccipital (Fig. 19A). The small endolymphatic foramen is located in the cavum capsularis, just lateral to the osseous common crus (*Oelrich, 1956*). It pierces the supraoccipital anteromedially and opens as a narrow foramen on the external medial surface of the supraoccipital.

## Basioccipital

The basioccipital forms the posteroventral floor of the braincase as well as the ventral and medial portions of the occipital condyle. The basioccipital contacts the otooccipital and prootic dorsally, the sphenoid anteriorly, and the atlas vertebra posteriorly (Figs. 18A and 18B). The basioccipital is anteriorly fused to the sphenoid, and there is no distinguishable suture visible on the ventral surface. The dorsal surface is concave and widest at the midpoint of the bone. The posterior half of the basioccipital tapers and projects posterodorsally. Small basal tubercles extend ventrolaterally into rounded ends, forming a deep groove on the ventral surface. Only the juvenile *Cyclura carinata* specimen UMMZ 117401 has paired foramina on the ventral surface, medial to the basal tubercles. The thin crista interfenestralis extends along the anterodorsal face of the basal tubercles, which also contributes the ventral border of the lateral aperture of the recessus scalae tympani (Fig. 19B).

## Sphenoid

The sphenoid forms the anteroventral portion of the braincase and features a parasphenoid process and a fused parasphenoid and basisphenoid (Figs. 18A and 19A). The sphenoid contacts the basioccipital posteriorly and the prootic dorsolaterally. The short alar processes project anteriorly from the dorsal surface, and slightly taper anteriorly into rounded ends (Fig. 19A). Medial to the alar processes, the small dorsum sella projects anteriorly and provides the posterodorsal border of the pituitary fossa (Fig. 20A). The pituitary fossa is

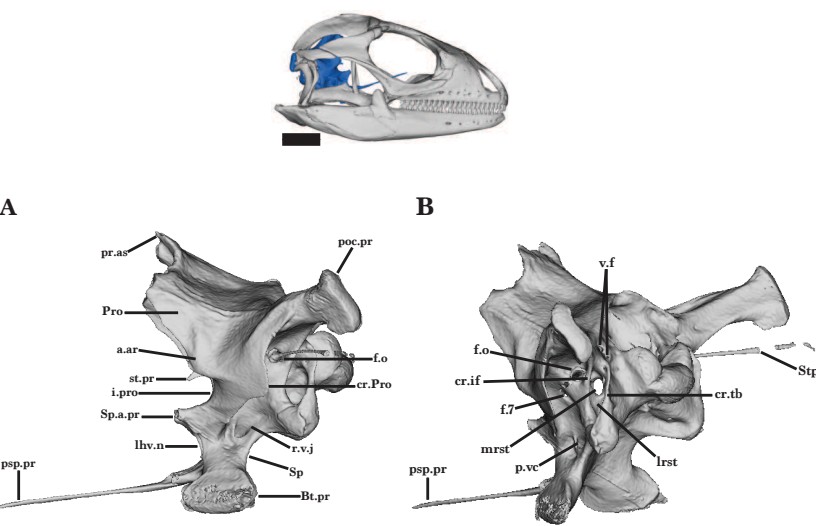

**Figure 20 General braincase 3.** General braincase of *Cyclura carinata* UF:Herp:32820 (ark:/87602/m4/M59620). (A) Lateral view, (B) posterolateral view. Scale bar = 10 mm.

deeply concave and opens anteriorly. The abducens foramen pierces the medial wall of the pituitary fossa and opens on the dorsal surface, just posterior to the crista sellaris (Figs. 20A and 20B). The anterior opening of the vidian canal is present posterolateral to the trabecula and ventral to the abducens foramen. The vidian canal opens posteriorly on the lateral face of the sphenoid and is medially bordered by the prootic (Fig. 19B). A thin medial septum on the posterior wall separates the internal carotid foramina and ends posterior to the sella turcica. The internal carotid foramen merges with the vidian canal, opening posteriorly.

Medial to the trabeculae, the long parasphenoid process projects anterodorsally and tapers to a point. A thin dorsal groove slightly widens anteriorly along the process (Fig. 18A). The parasphenoid elongates into the space medial to the pterygoids but does not articulate with the anterior palatine process of the pterygoid.

The basipterygoid process extends anterolaterally from the sphenoid and expands distally into rounded ends (Figs. 18B and 20A). The entire basipterygoid process slots into a large medial notch on the pterygoid, just anteroventral to the postepiterygoid groove. Between the alar process and the basipterygoid process, there is a small notch for the lateral head vein (internal jugular vein in *Porter & Witmer, 2015*).

## Prootic

The prootic is a paired, triradiate bone that forms the anterolateral region of the braincase. The prootic contacts the otooccipital posteriorly, the supraoccipital dorsally, the sphenoid anteroventrally, and the basioccipital posteroventrally (Figs. 18A and 19A). The anterior process of the prootic extends anterodorsally but does not extend anteriorly past the processus ascendens of the supraoccipital and does not contact the postparietal process of the parietal. The wide incisura prootica is formed along the dorsal margin of the anterior inferior process and is medioventral to the anterior ampular recess (Figs. 20B and 19A).

The ampullar recess is bulbous on the lateral surface of the prootic. A thin crest extends anteriorly along the medial surface of the prootic, terminating just posterior to the small supratrigeminal process (Fig. 20B).

A sharp crest runs posterodorsally from the ventral margin of the anterior inferior process to the posterior process of the prootic, forming the crista prootica (Fig. 19A). The crista prootica is expanded ventrally near its midpoint, forming a round, ventrally projecting lamina. The crista prootica continues onto the posterior process of the prootic, contacting the paraoccipital process of the otooccipital posteriorly. The posterior process contributes to the anterior region of the cavum capsularis, enclosing within it the anterior half of the horizontal semicircular canal. Along the ventral margin, the prootic contacts the alar process of the sphenoid to form the posterior opening of the vidian canal (Fig. 19B). The posterior opening expands onto the laterally exposed groove of the recessus vena jugularis. Posterodorsal to the recessus vena jugularis is the small facial foramen (VII), which is visible medially but obscured in lateral view by the crista prootica (Fig. 20B).

The medial face of the prootic has a deep pit, the acoustic recess, which is occupied by the medial openings of the large acoustic foramina and the facial foramen. The acoustic recess is located posterior to the incisura prootica. Dorsal to the facial foramen, the anterior acoustic foramen opens into the anterior ampullary recess, and the posterior acoustic foramen opens internally into the cavum capsularis. The paths of the anterior and horizontal semicircular canals are visible as prominent ridges on the lateral surface of the prootic (Fig. 19A). The anterior semicircular canal extends anterodorsally from the ampullar recess and opens at the articulation of the prootic and supraoccipital (*Evans, 2008*). The horizontal semicircular canal opens on the posterior process of the prootic. The prootic contacts the otooccipital posteromedially to form the anterior and lateral borders of the fenestra ovalis (Fig. 19B).

## Otooccipital

The otooccipital contributes to the posterodorsal portion of the braincase. The otooccipital contacts the basioccipital ventrally, the supraoccipital dorsally, and the prootic anterolaterally. The bone forms the posterior border of the cavum capsularis as well as the lateral borders of the foramen magnum. Ventral to the foramen magnum, the occipital condyle is composed of the medial basioccipital condyle and the triangular, dorsolateral knobs of the otooccipital (Fig. 18B). The otooccipital portions of the condyle are smaller than that of the basioccipital and are slightly mediolaterally compressed. Posteriorly, the two paraoccipital processes are large and robust and extend dorsolaterally into rounded ends. These posterior ends laterally articulate with the posteromedial facet of the supratemporal, the postparietal process, and the cephalic condyle of the quadrate. A well-developed ridge runs anteromedially along the paraoccipital process and merges with the thin crista interfenestralis. From the base of the paraoccipital process, the crista interfenestralis extends ventrally to the basal tubercles, separating the external lateral opening of the fenestra ovalis and lateral aperture for the recessus scala tympani (Fig. 19B).

Located dorsolateral to the crista tuberalis is the large fenestra ovalis, which is posteriorly bordered by both the otooccipital and prootic (Figs. 19A and 19B). The fenestra ovalis

opens posterolaterally and receives the expanded proximal end of the stapes. In anterior view, the fenestra ovalis pierces the bone medially and runs dorsolateral to the small lagenar recess. The thin medial crest of the lagenar recess bifurcates anteriorly, separating the recess from the posterior ampullar recess. Anterior to the posterior ampullar recess, a low ridge forms along the medial surface of the otooccipital as the utricular recess extends anteriorly. Ventral to the posterior ampullar recess is the perilymphatic foramen, which opens dorsal to the lagenar recess. The otooccipital contains the openings of the horizontal semicircular canal and the posterior semicircular canal at the articulations with the prootic and supraoccipital, respectively. Both semicircular canals connect anteriorly with the posterior ampullar recess.

The wide lateral aperture for the recessus scala tympani is oval in shape and is located ventral to the fenestra ovalis (Fig. 19B). It is bordered dorsolaterally by the crista interfenestralis and dorsomedially by the crista tuberalis, a distinct crest that extends ventrally onto the basal tubercle. There is a small foramen just ventral to the base of the paraoccipital process. The smaller, circular medial aperture of the recessus scala tympani is visible within the lateral aperture from a dorsolateral view. Dorsomedial to the lateral aperture and immediately lateral to the foramen magnum, there is a narrow pocket that is pierced by the vagus foramen (Fig. 19B). The lateral end of the vagus foramen is divided by a thin septum. Three hypoglossal foramina are aligned along the posteromedial surface, just ventral to the vagus foramen (Fig. 20B). The paraoccipital processes of the juvenile and MVZ Herp 81381 are not well-developed compared to UF Herp 32820, and terminate at the articulation with the cephalic condyle of the quadrate.

### Trachea

The trachea is a long, midline structure located in the posterior region of and posterior to the skull (Figs. 1 and 2). It is not a cranial bone, and is located just posterior to the posterior ends of the dentaries (Fig. 21). The trachea consists of a series of identical, cartilaginous rings that are evenly spaced and incomplete dorsally (*Oelrich, 1956*). Dorsal to the hyoid apparatus, the trachea extends posteroventrally past the atlas and axis vertebrae.

### Hyoid apparatus

The hyoid apparatus is located posteroventral to the skull and does not contact any other skeletal element (Fig. 2). The hyoid apparatus is characterized by symmetrical, gradually tapering ossifications that extend distally from the central basihyoid. The basihyoid is a triradiate structure consisting of the long processus lingualis and the paired second ceratobranchials.

The processus lingualis projects anteriorly from the basihyoid into the interpterygoid vacuity between the pterygoids, extending just anterior to the pterygoid teeth (Fig. 22B). The hyoid cornu is half the length of the processus lingualis and extends anterolaterally to the lateral processes of the pterygoids. Viewed laterally, the hyoid cornu slightly projects dorsally toward the dentaries (Fig. 22A). A long, thin anteromedial foramen is fully enclosed by the anterior end of the paired epihyals, which almost articulates with the hyoid cornu. The slender epiphyal extends posterolaterally past the atlas vertebra (*Paparella &*

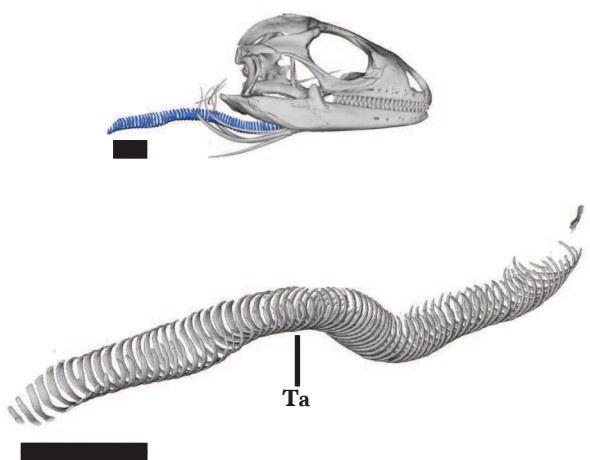

**Figure 21 Trachea.** Trachea of *Cyclura carinata* UF:Herp:32820 (ark:/87602/m4/M59620). In posterolateral view. Scale bar = 10 mm.

*Caldwell, 2021*). In lateral view, the epihyal projects posteroventrally and curves dorsally at a point ventral to the braincase (Fig. 22A). The posterior end tapers to a point and curves anterodorsally toward the posterior process of the articular. The paired first ceratobranchials possess knob-shaped ossifications at the anterior end that almost contact small anterolateral depressions on the second ceratobranchial. At a point ventral to the quadrate, the first ceratobranchial curves dorsally and gradually taper to a round end. The first epibranchial projects anterodorsally from the posterior end of the first ceratobranchial. The second ceratobranchial is fused to the basihyoid and extends posteriorly just ventral to the larynx. In lateral view, the second ceratobranchial projects posteroventrally along the entire length. The free epibranchial is located anteromedially to the first ceratobranchial and extends posterolaterally towards the first epibranchial (Fig. 22B). The hyoid apparatus of the juvenile is well-developed but is missing the left free epibranchial.

## Comparison of the skulls of *Cyclura carinata* and other species of iguanid

We examined traditionally prepared skeletons of *Cyclura cornuta* to provide a comparative benchmark of the anatomy of non-CT scanned specimens of *Cyclura*, and to provide some preliminary interspecific comparisons between *Cyclura carinata* and other members of the genus. Here we also present several interspecific comparisons within other iguanids.

### Dentition

Although there is variation in tooth crown morphology, typical *Cyclura* tooth crowns consist of a central cusp flanked by smaller cusps on the distal teeth (*Pregill, 1982*), and the mesial teeth are unicuspid. The Bahaman species *C. cychlura, C. carinata, C. rileyi,* and Cuban *C. nubila* share a tooth structure with two lateral cusps on either side of the central cusp (*Pregill, 1982*). The maxillae of *Cyclura* species generally have 26–28 tricuspid teeth (*Pregill, 1981*). The maxillae of *Brachylophus fasciatus* and *B. vitiensis* have plesiomorphic,

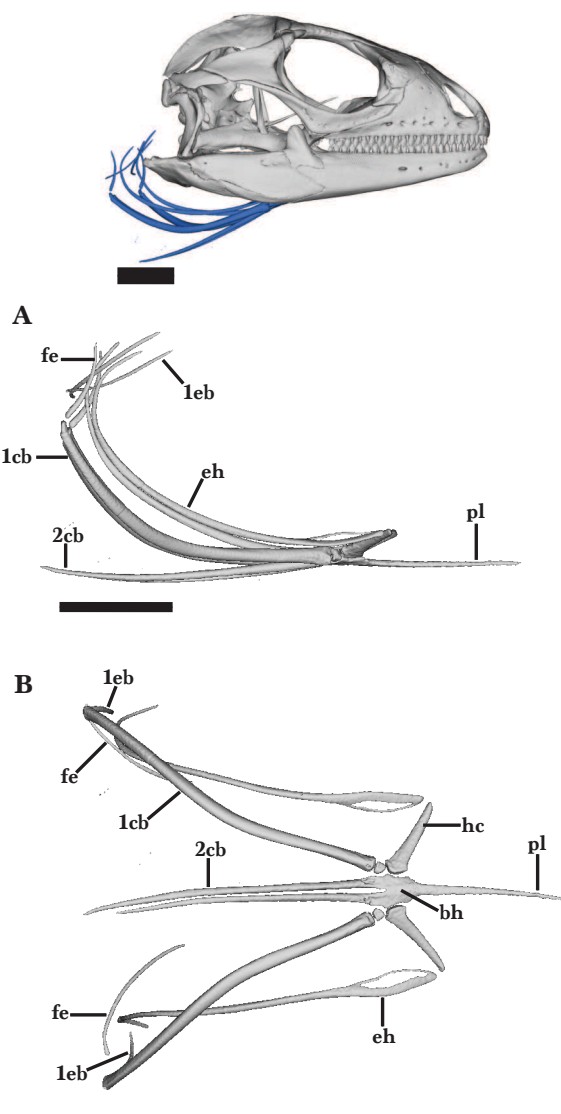

**Figure 22 Hyoid.** Hyoid of *Cyclura carinata* UF:Herp:32820 (ark:/87602/m4/M59620). (A) Lateral view, (B) ventral view. Scale bar = 10 mm.

tricuspid teeth; adult *Brachylophus* maxillae have around 18–21 teeth (*Pregill & Worthy, 2003*).

Both adult or near adult specimens of *Cyclura carinata* (Fig. 5B) and both examined specimens of *Cyclura cornuta* (MVZ 95982 and MVZ 95983) possess multicuspid posterior teeth on the dentary and maxilla (Figs. 23B and 23D). Tooth positions 5–21 on the adult *Cylura carinata* right dentary are multicuspid, with 3–5 clear cusps including a taller, well-developed middle cusp. The dentary of *Cyclura cornuta* MVZ 95982 has 26 tooth positions, with teeth increasing in size from positions 1–15 and decreasing from 15–26. The first seven dentary teeth are conical and unicuspid. Teeth 8–26 are multicuspid and all bear a taller middle cusp, similar to that of *Cyclura carinata*. However, on both specimens

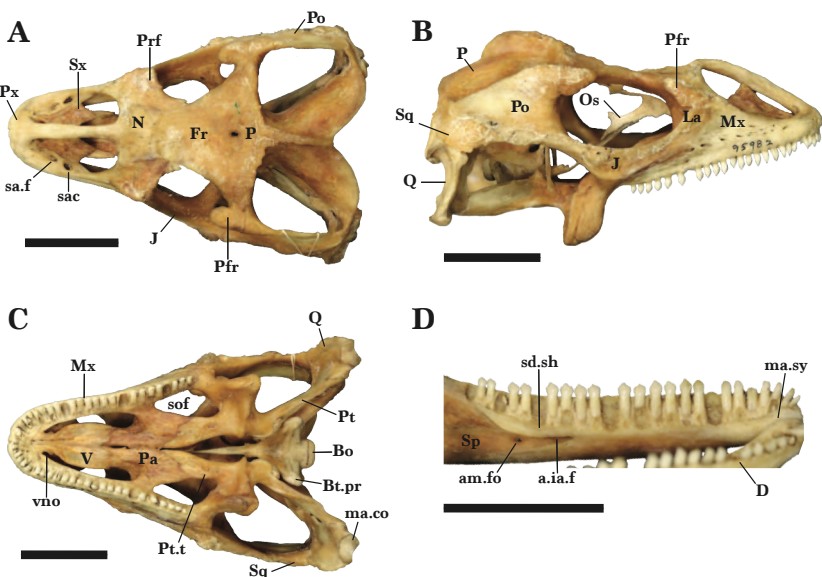

**Figure 23** **Cyclura cornuta entire skull.** Entire skull of Cyclura cornuta MVZ 95982. (A) Dorsal view, (B) lateral view, (C) ventral view, (D) left dentary in medial view. Scale bar = 3 cm.

of *Cyclura cornuta* there are 6–8 and potentially as many as 10 cusps on the distal dentary teeth, and 5–7 cusps on the distal maxillary teeth besides the last two teeth (Figs. 23B and 23D). Some of the smallest cusps on the specimens of *Cyclura cornuta* are difficult to distinguish, and so it is possible that there are more very small cusps on the distal teeth of the CT-scanned *Cylura carinata* that we were not able to observe from those data.

Like *Cylura carinata* and other *Cyclura* (*de Queiroz, 1987*), the premaxillary teeth of *Cyclura cornuta* are unicuspid. At least some of the premaxillary teeth have accessory cusps in all other iguanids besides some species of *Ctenosaura* (*de Queiroz, 1987*). Unicuspid premaxillary teeth may be an independently derived apomorphy in *Cylura* and *Ctenosaura*, or could be an apomorphy of the iguanid clade containing both of those species (see topology of *Zheng & Wiens, 2016*). Both skeletal specimens of *Cyclura cornuta* bear nine teeth on the premaxilla, and the specimens of *Cyclura carinata* have 7 or 8 teeth.

Iguanid species also differ with respect to pterygoid dentition, as pterygoid teeth can occur as a cluster, as in *Amblyrhynchus*, or in double rows, as in *Iguana* (*Pregill & Worthy, 2003*). Pterygoid teeth in *Amblyrhynchus* are arranged in a small pit or socket (*Paparella & Caldwell, 2021*), as with *Cyclura carinata* (Figs. 1B and 13).

## Premaxilla

The nares of *Cyclura cornuta* are narrow and ovular. In *Cyclura cornuta* MVZ 95983 and MVZ 95982, the septomaxilla anteriorly contacts the ventral surface of the premaxillary nasal process. *Cylura carinata* has a double pair of ethmoidal foramina, similar to the reported pair in *Ctenosaura*, as well as additional, smaller foramina below the ethmoidal foramina (*Oelrich, 1956*). These additional nutrient foramina are also documented in

*Amblyrhynchus* and do not exhibit symmetrical distribution on the surface of the maxilla (*Paparella & Caldwell, 2021*).

## Maxilla

The dorsal opening of the superior alveolar canal and the subnarial arterial foramen are present in all of the adult (and near adult) specimens of *Cyclura carinata* and *Cyclura cornuta* (Fig. 23A). Both of the subnarial arterial foramina on *Cyclura carinata* UF Herp 32820 and *Cyclura cornuta* MVZ 95983 are small. On *Cyclura cornuta* MVZ 95982, the foramen is oval in shape and larger than the dorsal opening of the superior alveolar canal. In *Amblyrhynchus,* the subnarial arterial foramen is absent and the premaxillary process (anterior process) of the maxilla is expanded relative to *Iguana*, similar to *Conolophus* (*Paparella & Caldwell, 2021*).

## Nasals

Both *Cyclura carinata* and *Cyclura cornuta* lack the large gap between the lateral margin of the nasal and the prefrontal that is present in the juvenile specimen. Randomly distributed foramina on the dorsal surface of the nasals are present in *Amblyrhynchus* and *Ctenosaura*.

## Frontal

Both examined specimens of *Cyclura cornuta* have relatively long, flat frontals (Fig. 23A). In lateral view, the dorsal surface of the skull is slightly depressed from the anterior end of the frontals to the anterior tip of the parietal crest.

## Parietal

The parietal descending process in *Cyclura cornuta* specimens MVZ 95983 and MVZ 95982 contacts the anterior process of the prootic, unlike in specimens of *Cyclura carinata*. This could potentially represent a discrepancy between CT and dried-skeletal data rather than an interspecific difference. A tall midline parietal crest is present on the dorsal surface of the parietal in *Cyclura cornuta* (Figs. 23A–23B), which is absent in examined *Cyclura carinata*. The parietal foramen is present in adult *Cyclura carinata* and *cornuta* specimens. The parietal foramen of *Cyclura cornuta* specimens is ovular (Fig. 23A). There are multiple foramina present on the dorsal surface of the parietal of *Cyclura carinata* UF Herp 32820 and MVZ Herp 81381 that are absent in *Cyclura cornuta* (Figs. 10A, 23A).

In *Amblyrhynchus, Brachylophus,* and *Iguana,* the parietal foramen (pineal foramen in *Paparella & Caldwell, 2021*; *Oelrich, 1956*) is located in the frontoparietal suture, which was hypothesized to be the plesiomorphic state in iguanids (*de Queiroz, 1987*; *Pregill & Worthy, 2003*). The parietal foramen of *Cyclura carinata* and *Ctenosaura* is also located at the frontoparietal suture, though is nearly contained by the frontal in UF Herp 32820. The parietal foramen of *Dipsosaurus* is located wholly within the frontal (*Oelrich, 1956*; *Norell & de Queiroz, 1991*).

## Postfrontal

In both skeletal specimens of *Cyclura cornuta*, the lateral process of the postfrontal is narrow and projects anteriorly (*Avery, 1970*; Fig. 23A). The postfrontal of *Amblyrhynchus* bears a large fossa behind the supraorbital boss, which is a feature absent in other iguanids

(*Paparella & Caldwell, 2021*). Unlike *Cyclura carinata* and *C. cornuta*, the postfrontal of *Amblyrhynchus* is pierced by small foramina on the medial and lateral walls (*Paparella & Caldwell, 2021*).

### Jugal

A bulbous projection on the lateral surface of the jugals is present at the midpoint of the jugals of both specimens of *Cyclura cornuta*. The jugals of specimens of *Cyclura carinata* have a smooth lateral surface and lack a lateral projection (Fig. 13A). The jugals of *Ctenosaura, Amblyrhynchus, C. cornuta,* and *C. cyclura,* have small foramina (suborbital foramina in *Oelrich, 1956*) along the lateral surface (*Paparella & Caldwell, 2021*; *Oelrich, 1956*). These foramina are present in young *Amblyrhynchus* individuals, but absent in the *C. carinata* juvenile.

### Pterygoid

The pterygoids of *Cyclura* and *Ctenosaura* both bifurcate anteriorly into two distinct anterior processes, the ventromedial and ventrolateral flanges (palatine process and transverse process in *Oelrich, 1956*). Unlike other iguanids, the *Amblyrhynchus* palatine process narrows to a point and bears a foramen (*Paparella & Caldwell, 2021*).

The pterygoid teeth are clumped together in a pit in *Cyclura carinata* and *Cyclura cornuta* MVZ 95982 (Fig. 23B) but are aligned in a row in *Cyclura cornuta* MVZ 95983. In *Cyclura carinata* MVZ Herp 81381, the pits for the pterygoid teeth are present, but teeth are not visible on either the segmented model or the CT slices. In *Amblyrhynchus*, there are three to five pterygoid teeth arranged in a small grove, whereas *Brachylophus fasciatus* and *B. vitiensis* have 8–10 teeth in a single row and that project slightly posterolaterally (*Paparella & Caldwell, 2021*; *Oelrich, 1956*). Pterygoid teeth are rarely present in *Dipsosaurus* and are absent in *Conolophus* (*Oelrich, 1956*).

### Epipterygoid

Unlike examined specimens *Cyclura carinata*, the epipterygoid of the dry skeletal specimens of *Cyclura cornuta* MVZ 95983 and MVZ 95982 contacts the parietal and and the elements are in much closer proximity than in the CT-scanned specimens. We suggest that shrinking of soft-tissue during skeletal preparation brings the parietal and epipterygoid into contact or close to contact.

### Prootic

*Cyclura carinata* bears a projection at the midbody of the crista prootica along the ventral margin (Fig. 19A). *Cyclura cornuta* MVZ 95983 also has this ventral projection, but it is less prominent than that of *Cyclura carinata*. *Cyclura cornuta* MVZ95982 does not possess the ventral protrusion. The crista prootica of *Amblyrhynchus* is straight and does not bear the ventral projection (ventral lappet), which is present in *Iguana* (*Paparella & Caldwell, 2021*). The anterior end of the prootic does not contact the epipterygoid in all examined specimens of *Cyclura*.

### Otooccipital

The lateral opening of the vagus foramen in *Cyclura carinata* UF Herp 32820 and MVZ 8138 and both *Cyclura cornuta* specimens appears to be split into two openings by a thin septum (Fig. 19B). This is not observed in other iguanids, as *Ctenosaura* and *Amblyrhynchus* have a vagus foramen without a septum.

### Hyoid apparatus

The elements of the hyoid apparatus are similar among *Cyclura carinata, Amblyrhynchus cristatus* (*Paparella & Caldwell, 2021*), and *Ctenosaura pectinata* (*Oelrich, 1956*).

## DISCUSSION

We provide the first, extensive insight into the skull, hyoid apparatus, and trachea of *Cyclura carinata*, establishing an anatomical framework for the skull of *Cyclura* and adding to the body of literature detailing the cranial osteology of extant iguanid lizards (*Bochaton et al., 2016*; *de Queiroz, 1987*; *Oelrich, 1956*; *Paparella & Caldwell, 2021*). Through the use of CT data and processing software, we constructed a digital, three-dimensional model of the skull and its disarticulated cranial elements. The model allowed observation of the articulations and fusions between bones and provided preliminary data on interspecific variation between *Cyclura carinata* and other *Cyclura* (*C. cornuta*). We also examined some differences between CT-scanned and traditionally prepared skeletal specimens and between *Cyclura* and other iguanids.

Morphological differences between adult and juvenile specimens, such as a large parietal opening or an underdeveloped premaxillary nasal process, indicate ontogenic traits that distinguish adult and juvenile *Cyclura carinata*. Ontogenetic increases in teeth cusps and size are observed between juvenile and adult *Cyclura carinata* (*de Queiroz, 1987*). While both the dentary and maxillary teeth of *Cyclura* specimens increase in size posteriorly, the teeth of the juvenile *Cyclura carinata* are not well-developed and many are unicuspid, whereas most teeth of the adult *Cyclura carinata* are multicuspid. The shape of the juvenile *Cyclura carinata* parietal is less defined compared to that of the adult specimen, and it exhibits a smooth dorsal surface and a large opening between the frontal and parietal bones that suggest ontogenetic modifications. The adult *Cyclura carinata* has a laterally compressed midbody and longer, well developed postparietal processes (*Bochaton et al., 2016*). Additionally, the adult specimen possesses a tall dorsal shelf and foramina across the dorsal surface that are not present on the juvenile specimen (Fig. 10A).

In the adult specimens, the heavily fused elements, such as the braincase, lacked clear suture lines between component elements and were not digitally disarticulated. Although the articular and surangular exhibited more prominent boundaries, the posterior end of the two bones appeared indistinguishably fused on the three-dimensional model and showed unclear divisions on the CT scans slices. As such, a small section of the posterior boundaries of the articular and surangular required approximation using the individual slices.

An expanded dataset of both CT data and skeletal specimens is essential for more extensive comparative analyses of intraspecific variation in cranial anatomy with *Cyclura*

*carinata*, and interspecific variation between species of *Cyclura*. Only a few specimens were examined here (one juvenile, one mature individual, and one individual that may have been near maturity), and so our reports of intraspecific and ontogenetic variation are preliminary. Possible ontogenetic differences, such as a medial gap between diverging nasals or absent foramina, were only observed on one juvenile *Cyclura carinata* specimen (UMMZ 117401) and are difficult to confirm without more juvenile specimens. Both examined specimens of *Cyclura cornuta* bear a well-developed, midline crest on the parietal that observed specimens of *Cyclura carinata* lack. A larger sampling will facilitate a deeper examination of any observed differences between and within species, allowing for a greater understanding of variation and a more comprehensive comparison and understanding of the cranial morphology of this genus.

Identification of fossils will be an important application of our study and future studies examining the cranial osteology of species of *Cyclura*, all of which are threatened with extinction (*Buckley et al., 2016*; *IUCN, 2023*). Documenting the osteological variation of extant *Cyclura* is key for identifying fossils based on morphology, and accurate fossil identifications permit a greater understanding of evolutionary and biogeographic history (*Bell, Gauthier & Bever, 2010*; *Parham et al., 2012*). Accurate identifications of fossil *Cyclura* may allow a better understanding of *Cyclura* speciation and past interactions with humans, such as how hunting by humans may have affected *Cyclura* distributions during the Holocene (*Woodley, 1980*). Broadly, an understanding of the biology of *Cyclura*, including osteology and past distributions inferred from the fossil record, will help to inform future conservation measures for this singular group of lizards.

## Data

All CT data were downloaded from or are now deposited at Morphosource.org. The CT dataset for the whole body of UMMZ 117401 (Media 000070945) was acquired from https://www.morphosource.org/concern/parent/000S21327/media/000070945. The CT dataset for the head of UF 32820 (Media 000059620) was acquired from https://www.morphosource.org/concern/media/000059620. The CT dataset for MVZ Herp 81381 was collected for this study and is now deposited at https://www.morphosource.org/concern/parent/000597395/media/000597398 (DOI https://doi.org/10.17602/M2/M597398).

### Institutional Abbreviations

| | |
|---|---|
| **UMMZ** | University of Michigan Museum of Zoology |
| **UF** | Florida Museum of Natural History (University of Florida, Gainesville) |
| **MVZ** | Museum of Vertebrate Zoology (Herpetology Collections) |

## ACKNOWLEDGEMENTS

We thank Jack Tseng (UC Berkeley) for scanning MVZ Herp 81381 and the Museum of Vertebrate Zoology for loaning that specimen and for other logistical support. We are grateful to David Ledesma for reviewing an earlier version of the manuscript. The editor and the reviewers Juan Diego Daza, Cristian Hernández-Morales, and Edward Stanley provided critical feedback which improved the manuscript substantially.

### Funding

This work was supported by NSF DBI (No. 2109461). The funders had no role in study design, data collection and analysis, decision to publish, or preparation of the manuscript.

### Grant Disclosures

The following grant information was disclosed by the authors:
NSF DBI: 2109461.

### Competing Interests

The authors declare there are no competing interests.

### Author Contributions

- Chloe Lai conceived and designed the experiments, performed the experiments, analyzed the data, prepared figures and/or tables, authored or reviewed drafts of the article, and approved the final draft.
- Simon G Scarpetta conceived and designed the experiments, performed the experiments, analyzed the data, prepared figures and/or tables, authored or reviewed drafts of the article, and approved the final draft.

### Data Availability

The data for UF:Herp:32820 are available at MorphoSource: https://www.morphosource.org/media/000059620. The scan creator is Edward Stanley at Florida Museum of Natural History, University of Florida, Division of Herpetology and is sponsored by NSF DBI # 1701714; oVert TCN.

The computed tomography data for the specimen that we scanned for this study (MVZ Herp 81381) are available at MorphoSource: DOI https://doi.org/10.17602/M2/M597398.

The computed tomography data for specimen UF:Herp:32820 is available at MorphoSource: https://www.morphosource.org/concern/media/000059620.

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
