# Peer review of "The skull of the Turks and Caicos rock iguana, Cyclura carinata (Squamata: Iguanidae)"

_PeerJ, doi:10.7717/peerj.17595_

## Round 0.1 · original submission · Major Revisions

All three reviewers suggested major revisions. Please, consider their comments and provide point-by-point response, when resubmitting new version of your manuscript

·

Basic reporting

The paper is very interesting and offers a detailed description of the skull of the Turks and Caicos rock iguana. This genus includes the largest iguanas of the world, so if information on this species is very valuable. I think the manuscript has the potential of becoming a very important paper, but requires some work before being in its final shape. I have made comments directly on the pdf. relevant literature is included, but there are some others that need to be included (see my notes).

Experimental design

The descriptions are very general. I recommend maybe for each bone specify the differneces seen when comparing the juvenile and the adult. I think a comparison with other Iguanas would increase the value of this paper, it doesn't; need to be extensive, but maybe this can produce some characters that can help diagnosing Cyclura. Besides, Thera are excellent and highly detailed descriptions of other species, so this can enhance the pair. I leave this merely as a recommendation, and the authorscan decide if this is out of the scope or no.

Validity of the findings

The paper has very interesting new data, and has been well illustrated. Since there are not available osteological descriptions of Cyclura, this paper will be useful for people working on Caribbean herpetology and paleontology.

Additional comments

There are some portions that can be removed, for example the trachea is visible and looks cool, but is an independent structure from the skull, so is better leave out. I recommend some slight improvement on the structure of the paper:
1) Each description can have a short sentence on the differences between the adult and juvenile,
2) A section with a broad comparisons with available descriptions of other iguanas (this is optional).
3) certainly remove the part of the trachea.

Juan D. Daza

·

Basic reporting

In general terms, this paper meets the standards of an anatomical description. The author presented a detailed (bone-by-bone) description and illustrations of the skull of Cyclura carinata. However, it is necessary to improve some aspects:

1. The introduction considers general aspects of the biology of Cyclura. Nevertheless, almost ignores the history of the study of the skeletal anatomy of Iguanians, please include it. Also, there are no mentions of the hyoid and trachea in the introduction. Please, explain why it can be interesting and comment on previous descriptions of these structures in other Iguanians (if available).

2. In M&M section there is an absence of the parameters you followed: references you use for the terminology, and the criteria to produce the figures.

3. The order of the figures does not follow the order of the description. Please re-order de figures or the description.

4. In the discussion, please use the available descriptions of other Igunians to compare to Cyclura. Are there any putative synapomorphies?

Experimental design

no comment

Validity of the findings

The paper is a valuable contribution to the study of Iguanian (and Squamata) anatomy. The results presented are important in many aspects as classification, ecology, systematics, etc.

·

Basic reporting

The manuscript uses CT data downloaded from Morphosource to produce a bone-by-bone assessment of Cyclura carinata. The paper is well written and has a good structure. The introduction is short but to the point, and highlights the benefits of these kinds of studies. I think the paper, with a few amendments and rearrangements will serve a range of communities and is appropriate for PeerJ.


The bone-by-bone descriptions, while accurate and well written, could benefit from more standardization in their structure. While this is a personal preference, I find it is most efficient to describe the location of the bone relative to the skull and surrounding bones first, then present a broad description of the bone shape, followed by highlighting specific features from anterior to posterior. That way the reader can rapidly jump to the information that they need.

The figures are well done, but replacing the labeled, monocromatic skull image with one that has all the bones colored differently would be useful to clearly delineate the bones. Jaimi Gray has an excellent standardized color palate for squamate cranial bones http://www.graysvertebrateanatomy.com/work/colorsofskullanatomy/ that would work well here. Finally, I find that volumetric renders always look better than isosurface renders, and 3DSlicer has recently added a “Colorize Volume Module” in the the Sandbox extension that lets users convert their separate materials into a series of isolated volumes that can be rendered with proportional transparency, which looks much better!

Experimental design

The Experimental design, as a basic inventory and description of the skull bones of Cyclura is well set out. One especially laudable aspect of this study is the inclusion of multiple individuals of the same species that represent an ontogenetic series of Cyclura, though it was a little disappointing how little information about ontogenetic variation in each bone was presented. Keeping the descriptions and ontogenetic variation separate is a stylistic choice, but given the amount of segmentation work that the authors have carried out, it would be worthwhile to include this information together. One way of incorporating the ontogenetic variation into the main body would be to outlie the structure for each bone, and then add a section of ontogenetic variation immediately afterwards, even if this just reads “juvenile and subadult squamosals appear similar to those of the adults” or the equivalent. While I know this is a lot of work, the most time-consuming part has been done (segmentation) and I do think that this would broaden the scope of the paper and set it apart from the other skeletal descriptions of similar species.

Validity of the findings

The bone-by-bone descriptions are well done for the most part and provide a good base for future studies in this area. The most immediate benefit of these kinds of studies is to paleontologists and ecologists, who often find isolated elements that need identification. The introduction could explore the uses of skeletal inventories a little to highlight the importance of the study. Are there Cyclura fossils? The (sometimes unreliable) PaleoDatabase page for Cyclura suggest that there are several across the Caribbean and at least one Holocene fossil on the Turks and Caicos (S. Jones O'Day. 2002. Late Prehistoric Lucayan occupation and subsistence on Middle Caicos Island, northern West Indies. Caribbean Journal of Science 38:1-10). there may be other potential uses for zooarcheologists and conservationists, which could be delved into.

Additional comments

Overall, the authors are to be commended for a detailed and well written body of work. My suggestions are offered to improve the scope and usability of the paper, but the foundation is already very good.

---

## Round 0.2 · accepted · Accept

This is an editorial acceptance; publication is dependent on authors meeting all journal policies and guidelines.

·

Basic reporting

The authors included most of the comments from the reviewers, and the manuscript has improved considerably. Now each element has a section of ontogenetic variation, which is a big improvement.

Experimental design

the methods used are adequate.

Validity of the findings

This is an important contribution to knowledge of this iconic species. I applaud the amount of effort and meticulous description of each bone.

Additional comments

Just a minor comment, I would like to thank the authors for their kind words in the akcnowldment section. Would it be possible to change my name to Juan Diego Daza, to distinguish me from my homonym Juan Manuel Daza, also a herpetologist!

Juan

·

Basic reporting

Dear authors,

In general terms, this paper meets the standards of an anatomical description. The authors presented a detailed (bone-by-bone) description and illustrations of the skull of Cyclura carinata. Most of my recommendations were well-addressed.

Now, I have an additional minor suggestion. This suggestion is merely aesthetic. Across the document, the scale bars are not standardized. The height of the scale bars is random. Obviously, The length of the scale bar varies. However, I recommend using scale bars with similar sizes; I'm not talking about the size they represent but the size they occupy in the figure. For example, the scale bars in Figure 3 are tiny squares, it does not look good. You could increase the length of the scale bar to represent 4 or 5 mm, it would benefit the appearance of the figures.

All the best
Cristian Hernandez

Experimental design

No comments

Validity of the findings

No comments

Additional comments

No comments

·

Basic reporting

The Authors have addressed my main concerns with the reporting in the paper by combining the juvenile and adult specimens in the bone-by-bone descriptions, standardizing the structure of the descriptions and expanding the introduction to include information about the impact of this study on paleontology. I would argue that the creation of one new figure with all the segmented bones given unique colors would not be that time consuming but it is acceptable as is.

Experimental design

As above, the Authors have addressed my concerns with the experimental design by combining the juvenile and adult specimens in the bone-by-bone descriptions.

Validity of the findings

The findings are valid and more accessible in this new version.